# An integrated analysis of the cancer genome atlas data discovers a hierarchical association structure across thirty three cancer types

**Khong-Loon Tiong**[1], **Nardnisa Sintupisut**[1], **Min-Chin Lin**[1,2], **Chih-Hung Cheng**[1], **Andrew Woolston**[1,3], **Chih-Hsu Lin**[1,4], **Mirrian Ho**[1], **Yu-Wei Lin**[1,5], **Sridevi Padakanti**[1], **Chen-Hsiang Yeang**[1]*

**1** Institute of Statistical Science, Academia Sinica, Section 2, Taipei, Taiwan, **2** Psomagen, Rockville, Maryland, United States of America, **3** Translational Cancer Immunotherapy & Genomics Lab, Barts Cancer Institute, Charterhouse Square, London, United Kingdom, **4** C3.ai, Redwood City, California, United States of America, **5** AiLife Diagnostics, Pearland, Texas, United States of America

* chyeang@stat.sinica.edu.tw

**Data Availability Statement:** We place the IHAS inference and validation results in Supplementary

## Abstract

Cancer cells harbor molecular alterations at all levels of information processing. Genomic/epigenomic and transcriptomic alterations are inter-related between genes, within and across cancer types and may affect clinical phenotypes. Despite the abundant prior studies of integrating cancer multi-omics data, none of them organizes these associations in a hierarchical structure and validates the discoveries in extensive external data. We infer this Integrated Hierarchical Association Structure (IHAS) from the complete data of The Cancer Genome Atlas (TCGA) and compile a compendium of cancer multi-omics associations. Intriguingly, diverse alterations on genomes/epigenomes from multiple cancer types impact transcriptions of 18 Gene Groups. Half of them are further reduced to three Meta Gene Groups enriched with (1) immune and inflammatory responses, (2) embryonic development and neurogenesis, (3) cell cycle process and DNA repair. Over 80% of the clinical/molecular phenotypes reported in TCGA are aligned with the combinatorial expressions of Meta Gene Groups, Gene Groups, and other IHAS subunits. Furthermore, IHAS derived from TCGA is validated in more than 300 external datasets including multi-omics measurements and cellular responses upon drug treatments and gene perturbations in tumors, cancer cell lines, and normal tissues. To sum up, IHAS stratifies patients in terms of molecular signatures of its subunits, selects targeted genes or drugs for precision cancer therapy, and demonstrates that associations between survival times and transcriptional biomarkers may vary with cancer types. These rich information is critical for diagnosis and treatments of cancers.

## Author summary

Cancer cells harbor molecular alterations at all levels of information processing. These alterations are inter-related and affect the clinical traits in a complicated manner. We infer the associations of these molecular and clinical features and construct an Integrated

Data organized as hierarchical Webpages and deposited in the Synapse database.

**Funding:** CHY is supported by the Career Development Award of Academia Sinica (grant CDA-104-M04) and by the Ministry of Science & Technology in Taiwan (grants 108-2118-M-001-001-MY2; 107-21180M-001-007-; and 106-2118-M-001-012-). KLT, NS, MCL, CHC, AW, CHL, YWL, and SP received salary from the Career Development Award of Academia Sinica (grant number CDA-104-M04). The funders had no role in study design, data collection and analysis, decision to publish, or preparation of the manuscript.

**Competing interests:** There is no competing interest.

Hierarchical Association Structure (IHAS) from The Cancer Genome Atlas (TCGA) data. IHAS provides a unique contribution in cancer omics as it (1) represents complicated associations in a hierarchical structure and presents varying levels-of-details views from a single gene in a specific cancer type to groups of genes across multiple cancer types, (2) performs both vertical (across multiple types of assays) and horizontal (across multiple cancer types) data integrations, (3) incorporates a large-scale biological knowledge base in the model, (4) validates the inferred associations in over 300 external datasets. In the long term, IHAS can illuminate the universal and idiosyncratic aspects of cancer omics data, give new insights about diagnosis, and provide guidance for targeted cancer therapies for precision medicine.

## Introduction

Cancer cells harbor alterations at all levels of information processing in the central dogma. In a nutshell, there are driver and passenger alterations with and without conferring phenotypic consequences [1], and driver alterations perturb *hallmark* processes covering major cellular functions of all tissue types [2]. Molecular alterations and oncological phenotypes are inter-related in a complicated manner. For instance, tumors derived from distinct tissues exhibit strong diversity [3], and tumor genomes are fast evolving and quickly adaptive to resist state-of-the-art treatments [4]. Comprehensive knowledge of these relations is crucial for curing cancer, as patterns arising from these relations are indicators for prognostic outcomes, drug responses, and selection of targeted treatments and manipulations/perturbations of these relations can effectively shrink or eliminate tumors.

Despite its importance, this comprehensive knowledge is still at far horizon due to complexities in at least three aspects. First, from an epistemological perspective these relations exist at statistical (associations from observational data), causal (consequences of intervention), and mechanistic (realization through molecular interactions and biochemical reactions) levels. Second, alterations occur at all information processing machinery including genomic, epigenomic, transcriptomic, proteomic, metabolomic, and phenotypic levels. Third, some relations are universal across all cancer types, but others can be highly specific to certain cancer types. Nevertheless, there are already two major catalogs of molecular alterations in diverse cancers. The Cancer Genome Atlas (TCGA) generates seven types of omics data and rich clinical information of over 11000 patients across 33 cancer types [5–16]. The International Cancer Genome Consortium (ICGC) generates the multi-omics data covering more cancer types and the whole genomes [17–20]. These databases provide the comprehensive molecular alteration "parts lists" underlying the network of their relations, yet building the entire network remains an unsolved and daunting challenge.

Many of these relations comply with the information flow according to the central dogma. Molecular aberrations on the genome (such as sequence mutations and copy number variations or CNVs) and epigenome (such as DNA methylations and chromatin modifications) mis-regulate abundance in the transcriptome and proteome, which affects clinical and physiological phenotypes (such as proliferation, metastasis, and treatment response). These causal links impact all aspects of cancer research including fundamental biology (e.g., which metabolic pathways are activated upon TP53 mutations), diagnosis (e.g., what are the CNV hallmarks for breast cancer basal tumors), and treatment (e.g., what are the targeted drugs to treat tumors of heightened leukocyte infiltration level). The high-level goal of this work is to reconstruct these causal links in all 33 cancer types of TCGA data. Specifically, we have (1)

developed and implemented a data integration framework to infer associations between molecular alterations on genomes/epigenomes and transcriptomes, and between transcriptomes and clinical/molecular phenotypes, (2) provided a compendium of these inferred associations covering 7 omics data types and across 33 cancer types, (3) organized these associations in a hierarchical structure allowing investigations at multiple levels of details, which we termed as the Integrated Hierarchical Association Structure (IHAS), (4) validated IHAS in a wide range of external datasets.

A large number of prior studies have inferred the relations of genes or molecular alterations. Many of them integrate the information from multi-omics data probing the same cohort or cancer type (*vertical integration* according to [21], see review articles such as [22–26]), or combine the single-omics data probing multiple cancer types (*horizontal integration*). Vertical integration methods are widely employed to cluster patients or samples [27–31], predict clinical outcomes [32–34], identify biomarkers or driver genes [35–38], and infer pathway or subnetwork activities [39–43]. Horizontal integration is typically achieved by *pan-cancer* studies that analyze the joint data concatenated from all cancer types (e.g., [11–16,44–49]). Another approach, termed as *meta analysis*, analyzes the data of each cohort separately and combines the analysis results afterward ([50–53]). In the context of multi-omics pan-cancer data integration, a number of recent methods have been successfully employed to analyze TCGA or other cancer data (e.g., [12,54–57]). Several approaches are also based on the central dogma information flow (e.g., [35,36,39–41,56]).

Despite rapid progress in cancer omics data integration, no prior studies have simultaneously achieved the four goals accomplished in our work. Most aforementioned studies conducted proof-of-concept investigations on the data of selected cancer types rather than the entire TCGA data. The pan-cancer papers published on *Cell Issue 173* (2018) tackled diverse aspects of TCGA data but did not cover genome/epigenome-transcriptome associations. All previous approaches tackling genome/epigenome-transcriptome associations reported a flat modular structure (one upstream regulator influences multiple downstream targets), but did not attempt to find a concise representation of the massive structure when modules may possess complicated relations within and across cancer types. Furthermore, no prior studies validated the inferred models in a wide range ($>$300) of external datasets from both tumors and normal tissues. Therefore, IHAS provides unique contributions in cancer omics data integration.

## Results

### Overview of integrated analysis and validation on pan-cancer omics data

**An example of integrated hierarchical associations in breast cancer.** We elucidate IHAS with an example of TCGA breast cancer data (BRCA) and the data of other cancer types in Fig 1. In panel A, E2F1 (located on chr20q) mRNA expression profile (the bottom row) is associated with 22 features of molecular alterations (the remaining rows). They constitute an *Association Model* of E2F1 as the *target* and the remaining features as the *effectors*. One effector is chr8q CNV (the 4th row marked by yellow boundaries), which is termed trans-acting CNV since it is located at a different chromosome from the target gene. In panel B, chr8q CNV variations (the top row marked by a cyan boundary) is associated with over 1000 target gene expressions including E2F1 (the yellow line). They constitute an *Association Module* of chr8q CNV. In panel C, chr8q CNV (the row marked by yellow boundaries) co-associates with dozens of other molecular alterations (the rows above the cyan boundary) with thousands of target genes (the rows below the cyan boundary). Those effectors include trans-acting CNVs, mutations, DNA methylations, mircorNA expressions, and protein phosphorylations. The

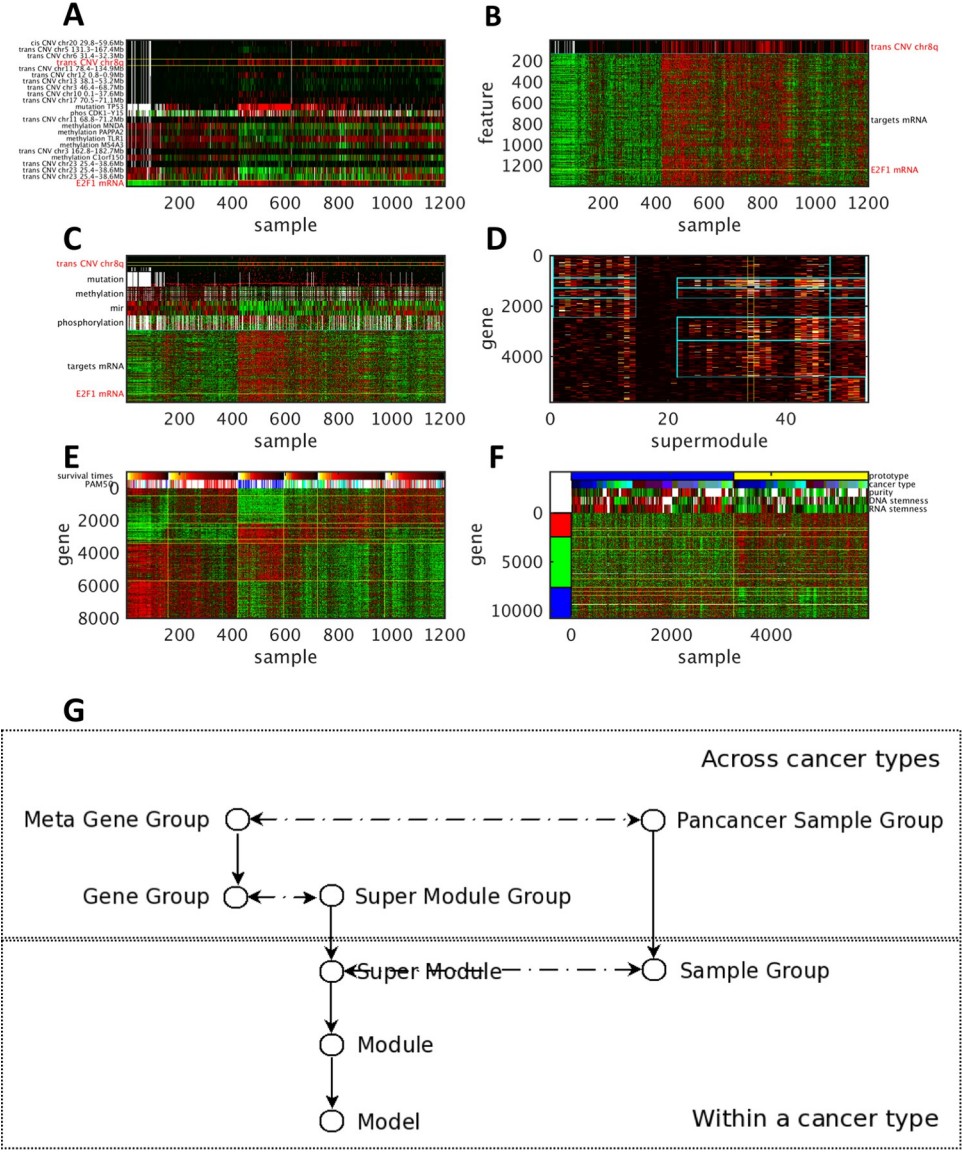

**Fig 1. An illustration of integrated hierarchical association structure from BRCA and other cancer types of TCGA data. A**: The Association Model of E2F1 as the target gene in TCGA BRCA data. The effectors and target of the Model are annotated. The bottom row displays its target gene expressions across 1201 samples, and all the other rows display the effector variations of the Model. One effector–chr8q CNV–is marked by yellow boundaries. The effector variations and target gene expression values are normalized in [0,1]. Red and green entries indicate high (1) and low (0) values respectively, and dark entries indicate the intermediate values (0.5). **B**: The Association Module of chr8q CNV as the effector. The top row displays chr8q CNV, and all the other rows display the target gene expressions. E2F1 gene expressions are marked by a yellow line. **C**: The effector variations (upper rows) and target gene expressions (lower rows) of a BRCA Super Module. Effectors and targets are separated by a cyan boundary. Effectors are sorted top-down by the types of associations in the following order: trans-acting CNVs, mutations, DNA methylations, microRNA expressions, and protein phosphorylations. Chr8q CNV and E2F1 gene expressions are marked by yellow boundaries. **D**: The membership occurrence matrix of 7 Gene Groups in 4 Super Module Groups including the BRCA Super Module in panel **C**. The heat color of each entry reflects the number of Modules that contain a gene (a row) and belong to a Super Module (a column). The column of the BRCA Super Module in panel **C** is marked by yellow boundaries. The occurrence of E2F1 in the BRCA Super Module is marked by a blue bar. A rectangular patch marked by cyan boundaries indicates the over-representation of a Gene Group in a Super Module Group. **E**: The combinatorial expressions of Super Modules and Sample Groups in BRCA data and their alignments with clinical phenotypes. Genes are included if they appear in at least 10 Association Modules in any Super Module. Eight Super Modules and six Sample Groups are demarcated by yellow horizontal and vertical lines respectively. The sorted samples are aligned with four PAM50 subtypes of breast cancer (blue: basal-like, green: HER2-enriched, red: luminal A cyan: luminal B)

and survival times (brighter heat colors denote longer survival times). Samples within each Sample Group are sorted by survival times in a descending order, and the boundaries of the five-year (1800 days) survival time are marked by green vertical lines. **F**: The combinatorial expressions of 10 Gene Groups belonging to 3 Meta Gene Groups and 2 Pan-cancer Sample Groups across all cancer types and their alignments with several pan-cancer phenotypes. The Gene Groups are demarcated by yellow horizontal lines and aggregated to three Meta Gene Groups marked by red, green and blue bars respectively. The two Pan-cancer Sample Groups are separated by a yellow vertical line. Pan-cancer phenotypes of RNA and DNA stemness, sample purity, cancer types, and prototypes (Pan-cancer Sample Groups) of the samples are displayed. **G**: An overview of relations between subunits in IHAS. Vertical unidirectional lines indicate inclusion relations; for instance, a Module is a collection of Models, and a Super Module is a collection of Modules. Horizontal bidirectional dashed lines indicate that subunits at the same level possess certain combinatorial relations; for instance, Super Modules and Sample Groups of the same cancer type form the combinatorial expression patterns like panel **E**, and Super Module Groups and Gene Groups constitute a membership occurrence matrix like panel **D**. Subunits within and across cancer types are placed at the lower and upper parts of the diagram.

effectors and targets constitute a *Super Module* in BRCA. The Super Modules across multiple cancer types are clustered to form *Super Module Groups* in terms of the overlap ratios of their target genes. Each Super Module Group is enriched with a combination of several *Gene Groups*. The heat map in panel D displays the membership occurrence matrix of genes (rows) in selected Super Modules (columns), where the entry brightness denotes the frequency each gene appears in each Super Module. The BRCA Super Module in panel C (the column marked by yellow boundaries in panel D) belongs to a Super Module Group in the middle (between 20 and 50), and this Super Module Group is enriched with four Gene Groups (four patches marked by cyan boundaries). In panel E, samples within BRCA data are clustered to form *Sample Groups* according to their combinatorial expression patterns of Super Modules. The samples sorted by Sample Groups are also aligned with the PAM50 subtypes of breast cancer and survival times. Based on functional enrichment and combinatorial expressions, Gene Groups are further clustered to 3 *Meta Gene Groups*, and Sample Groups across cancer types are further clustered to 8 *Pan-cancer Sample Groups*. Panel F displays the combinatorial expressions of 3 Meta Gene Groups and 2 Pan-cancer Sample Groups. The samples in two Pan-cancer Sample Groups (1 and 6) possess the opposite expression patterns of the three Meta Gene Groups, exhibit very different values in several pan-cancer phenotypes including sample purity, DNA and RNA stemness, and are from diverse cancer types. Panel G summarizes the relations of these IHAS subunits. There are three chains of inclusion relations. Association Models are hierarchically organized into Modules, Super Modules, and Super Module Groups. Genes are hierarchically organized into Gene Groups and Meta Gene Groups. Samples are hierarchically organized into Sample Groups and Pan-cancer Sample Groups. Furthermore, some subunits at the same level possess certain combinatorial relations. Super Modules and Sample Groups within the same cancer type form the combinatorial expression patterns pertaining to specific functional categories and subtypes (e.g., differential expressions of genes pertaining to estrogen receptors, immune responses and cell cycle control in the four PAM50 subtypes of breast cancer in panel E). Super Module Groups and Gene Groups constitute the membership occurrence matrix (panel D). Meta Gene Groups and Pan-cancer Sample Groups form global combinatorial expressions across multiple cancer types (panel F).

**The IHAS inference and validation framework.** Fig 2 illustrates the integrated analysis and validation framework of pan-cancer omics data. We downloaded the TCGA data covering 33 cancer types (Table 1) and 7 types of molecular features–mRNA expressions, CNVs, gene mutations, DNA methylations, microRNA expressions, protein expressions and phosphorylations, SNPs–as well as various clinical features such as survival/censoring times and molecular subtypes. Within each cancer type, we transformed the omics data into simplified and standard formats (the top row in Fig 2). The association subunits such as Association Models and Module and Super Modules were hierarchically inferred from the transformed data of each

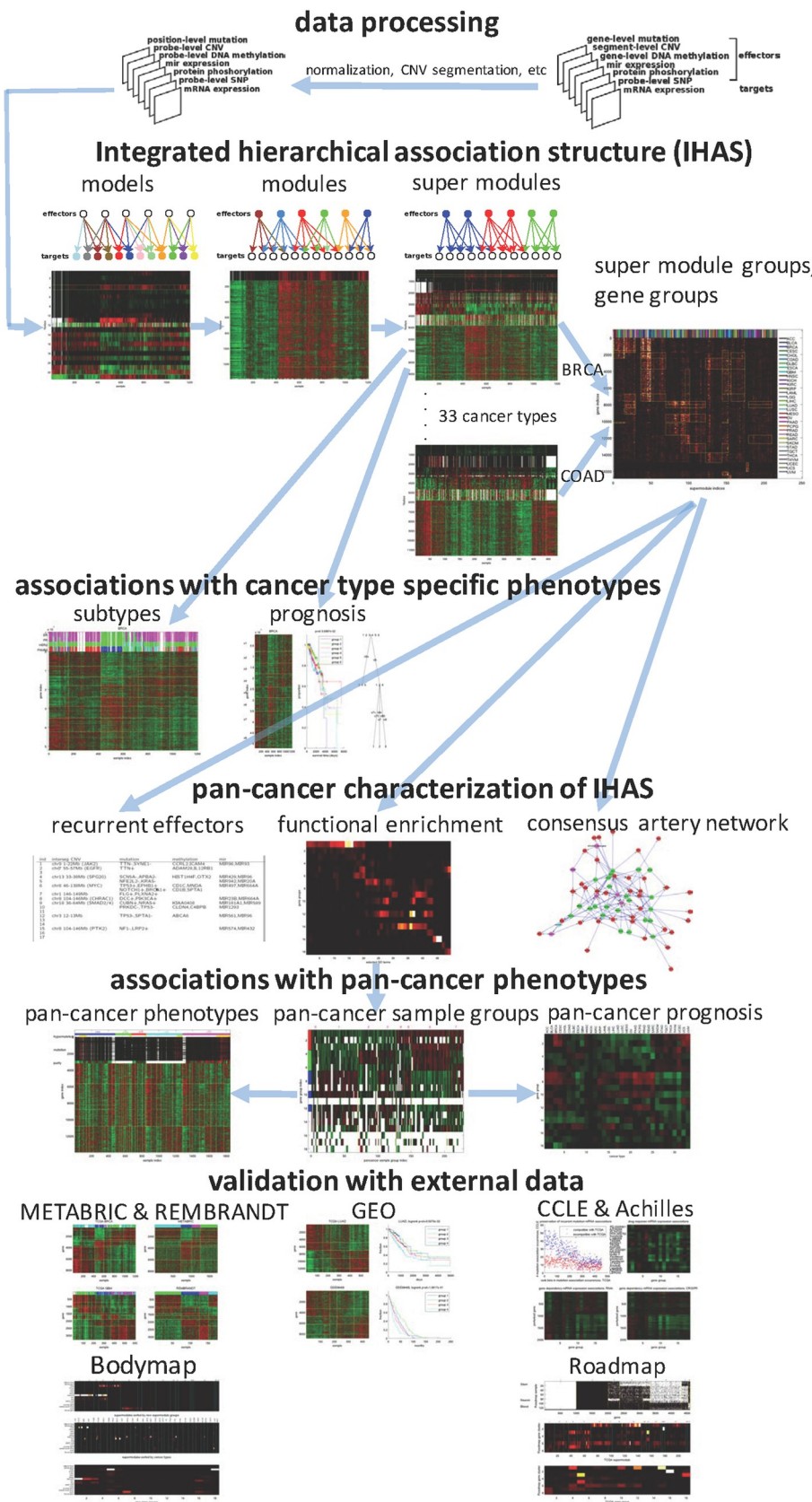

**Fig 2. Overview of the integrated analysis and validation framework of pan-cancer omics data.** For each cancer type, multiple types of omics data are converted into the same format (the first row). The hierarchical association structure is inferred from the processed data and includes Association Models, Association Modules, and Super Modules within cancer types and Super Module Groups and Gene Groups across cancer types (the second row). Effectors or targets of the same color in three subgraphs belong to the same Association Models, Association Modules, and Super Modules. We characterize three functional aspects of the hierarchical association structure: Recurrent Effectors that frequently appear in Super Module Groups, functional enrichment of Gene Groups, and the Artery Network that explains many (effector,target) associations in the molecular interaction network (the fourth row). Samples are grouped by combinatorial expression patterns of Super Modules and three Meta Gene Groups within and across cancer types, and are associated with survival outcomes and clinical phenotypes (the third and fifth rows). Finally, the hierarchical association structure is extensively validated in external datasets (the sixth and seventh rows) including the multi-omics data of breast cancer and glioblastoma tumors (METABRIC and REMBRANDT), 294 cancer transcriptomic datasets from GEO, multi-omics cancer cell line data (CCLE and Achilles), transcriptomic and epigenomic data of normal tissues (Bodymap and Roadmap). Blue arrows indicate prerequisite relations of steps in the framework. For instance, the three steps of pan-cancer characterization of IHAS all require Super Module Groups and Gene Groups.

cancer type, and Super Modules across cancer types were integrated to form Super Module Groups and Gene Groups (the second row). Within each cancer type, we also clustered samples into Sample Groups in terms of the expression patterns of Association Modules, and aligned Sample Groups with phenotypes such as molecular subtypes and survival times (the third row). We then characterized IHAS across cancer types with three analyses (the fourth row): functional enrichment of Gene Groups and Super Modules, identification of recurrent effectors in each Super Module Group, and construction of the *Artery Network* of biomolecular interactions which putatively relay the effector-target associations. Functional characterization further reduced Gene Groups into three *Meta Gene Groups* and Sample Groups into eight *Pan-cancer Sample Groups*, and we aligned them with pan-cancer phenotypes and prognostic outcomes (the fifth row). Finally, we validated IHAS with external datasets from five sources (the sixth and seventh rows). Summary explanations of IHAS terms are shown in Table 2, and S1 Fig depicts the architecture and information flows of the IHAS inference machine. For clarity we briefly introduce key IHAS components and report the detailed descriptions of this framework in Methods and S1 Text.

**Table 1. 33 cancer types in TCGA data and their sample sizes.**

| cancer type | description | # samples | cancer type | description | # samples |
|---|---|---|---|---|---|
| ACC | Adrenocortical carcinoma | 185 | LUSC | Lung squamous cell carcinoma | 1079 |
| BLCA | Bladder urothelial carcinoma | 843 | MESO | Mesothelioma | 175 |
| BRCA | Breast invasive carcinoma | 2281 | OV | Ovarian serous cystadenocarcinoma | 1213 |
| CESC | Cervical squamous cell carcinoma and endocervical adrenocarcinoma | 621 | PAAD | Pancreatic adenocarcinoma | 379 |
| CHOL | Cholangiocarcinoma | 116 | PCPG | Pheochromocytoma and paraganglioma | 367 |
| COAD | Colon adrenocarcinoma | 982 | PRAD | Prostate adenocarcinoma | 1061 |
| DLBC | Lymphoid neoplasm diffuse large B-cell lymphoma | 108 | READ | Rectum adenocarcinoma | 355 |
| ESCA | Esophageal carcinoma | 379 | SARC | Sarcoma | 532 |
| GBM | Glioblastoma multiforme | 1172 | SKCM | Skin cutaneous melanoma | 952 |
| HNSC | Head and neck squamous cell carcinoma | 1125 | STAD | Stomach adenocarcinoma | 942 |
| KICH | Kidney chromophobe | 227 | TGCT | Testicular germ cell tumors | 307 |
| KIRC | Kidney renal clear cell carcinoma | 1101 | THCA | Thyroid carcinoma | 1055 |
| KIRP | Kidney renal papillary cell carcinoma | 616 | THYM | Thymoma | 251 |
| LAML | Acute myeloid leukemia | 401 | UCEC | Uterine corpus endometrial carcinoma | 1126 |
| LGG | Brain lower grade glioma | 1048 | UCS | Uterine carcinosarcoma | 115 |
| LIHC | Liver hepatocellular carcinoma | 797 | UVM | Uveal melanoma | 161 |
| LUAD | Lung adenocarcinoma | 1287 | | | |

**Table 2. Summary description of subunits in IHAS.**

| Term | Description |
| --- | --- |
| Association Model | Effector molecular alterations associated with the expressions of one target gene |
| Association Module | Target genes shared with one common effector |
| Super Module | Association Modules in the same cancer type with substantially overlapped target genes |
| Sample Group | Samples in the same cancer type with shared combinatorial expression patterns of Super Modules |
| Super Module Group | Super Modules across cancer types with substantially shared target genes |
| Gene Group | Genes co-occurred as targets in one or multiple Super Module Groups |
| Meta Gene Group | Gene Groups enriched with one of three functions: immune response, development and cell cycle |
| Gene Set | Genes with a known function or co-expressed in a dataset from the MsigDB database |
| Recurrent Effector | Effectors with significant occurrence frequencies in a Super Module Group |
| Artery Network | Subnetwork of molecular interactions traversed by connecting paths of many association pairs |
| Consensus Artery Network | Common portion of Artery Networks across multiple cancer types |
| Pan-cancer Sample Group | Sample Groups across cancer types with shared combinatorial expression patterns of Meta Gene Groups |
| Patient Group | Patients derived from one or multiple Sample Groups |
| Cancer Subtype | Subtypes defined by genomic/transcriptomic signatures or clinical traits, such as PAM50 subtypes in breast cancer and CMS subtypes in colorectal cancer |
| Achilles Gene Cluster | Perturbed genes with similar correlation coefficients between their dependency responses in Achilles data and mRNA expressions of Gene Groups in CCLE data |
| Bodymap Tissue-Specific Gene Set | Genes uniquely expressed in one normal tissue in Bodymap data |
| Roadmap Gene Cluster | Genes with similar binary epigenomic states in Roadmap data |

**Association models.** S2A Fig shows the schematic diagram of the relations between molecular alterations. Beyond the central dogma links we also added associations from non-transcriptomic molecular alterations (mircoRNA expressions and protein phosphorylations) to mRNA expression variations. An Association Model fits the mRNA expressions of one target gene with one or multiple effector molecular alterations. We considered seven types of effectors: (1) Cis-acting CNV where a chromosomal segment encompasses or approximates the target gene, (2) Trans-acting CNV where a chromosomal segment is located on a different chromosome or chromosomal arm from the target gene, (3) Mutation of a gene, (4) Aggregate DNA methylation of a gene, (5) MicroRNA expression, (6) Phosphorylation of an amino acid residue in a selected protein, (7) SNP of a locus. The conditional density of the target gene expression $y$ (dependent variable) given the effectors $\boldsymbol{x}$ (independent variables) is specified by an exponential family distribution.

$$p(y|\boldsymbol{x}) = \frac{1}{Z(f(\boldsymbol{x}))} e^{\sum_i \lambda_i f_i(x_i) y}, \lambda_i \geq 0 \forall i. \tag{1}$$

where $f_i(x_i)$ is the feature function of the $i^{\text{th}}$ component of $\boldsymbol{x}$ relating the effector state values ($x_i$) to the target state values ($y$). For instance, $f_i(x_i) = x_i$ and $f_i(x_i) = -x_i$ denote that the effector activates or represses the target gene expression respectively. $\lambda_i$'s are free parameters of the model. $Z(f(\boldsymbol{x}))$ is a normalization constant. $\boldsymbol{x}$ and $y$ are all discrete random variables. To preserve the information about the continuous values of gene expressions or CNVs, we proposed a *probabilistic quantization* procedure ([40] and S1 Text Section 2.1.1) by treating the continuous data (such as mRNA expressions) as noisy measurements of discrete hidden variables

(such as the states of up/down regulation or no change) and relating the posterior probabilities of the hidden states with the measurement values. This model resembles logistic regression in terms of the formulation and parameter inference, but has a superior explanatory power as it can handle continuous dependent variables and nonlinear feature functions.

Building Association Models for the mRNA data of about twenty thousand genes from hundreds of thousands of putative effectors across 33 cancer types is challenging in three aspects. Statistically, spurious associations likely arise given the enormous number of candidate effector-target pairs and relatively small sample sizes. Biologically, associations of the same statistical strength may confer diverse relevance to the mechanistic/causal links. Computationally, finding Association Models from such a large number of covariates is time-consuming. We proposed several methods to address these challenges as illustrated in S2B Fig. First, we developed a stepwise regression-like model selection algorithm to build the association model. In each step, a candidate feature ($M_1'$) is added to the current model ($M_1$) only when the augmented model ($M_2$) significantly better fits the data than $M_1$ and $M_1'$ in terms of $\chi^2$ and permutation p-values. Second, candidate effectors are prioritized in terms of relevance and directness in a network of known biomolecular interactions. Effectors with shorter distances to the target have higher priorities since they are more likely to affect the target. Effectors with the same distance are further ordered by their types. Third, several computational techniques and parallel computation are employed to speed up association model inference.

**Association modules.**   Association Models are grouped by effectors to form Association Modules. For each effector (e.g., chr8q CNV), we collected the target genes whose Association Models included this effector and the corresponding feature functions had the same direction (positive or negative associations). Each Association Module consists of one effector and multiple targets sharing the common effector. We categorized Association Modules by seven types of effectors and two directions of associations. Cis-acting CNV modules possess only positive associations, and DNA methylation and microRNA expression modules possess only negative associations. For modules with trans-acting CNV and SNP effectors, we also required the presence of *regulators* mediating associations from effectors to targets. There are totally 11 types of Association Modules. In the example of Fig 1, chr8q amplifications likely up-regulate downstream targets on other chromosomes via a regulator MYC [58].

**Super Modules and Sample Groups.**   Within each cancer type, the Association Modules containing target genes with similar expression profiles form a Super Module. Each Super Module hence comprises the union of the effectors and targets of the member Association Modules. Likewise, samples sharing similar expression profiles of the Association Module target genes form a Sample Group. We employed a spectral clustering algorithm to simultaneously partition the mean expression data of Association Module targets into Super Modules and Sample Groups.

**Super Module Groups and Gene Groups.**   The membership relations of genes in Super Modules across all cancer types constitute a discrete-value matrix $M$ of genes (rows) and Super Modules (columns), where an entry $M_{ij}$ denotes the number of Association Modules which contain target gene $i$ and belong to Super Module $j$. Columns (Super Modules) and rows (genes) of $M$ form Super Module Groups and Gene Groups by a bi-clustering algorithm. First, we applied hierarchical clustering to form Super Module Groups according to the Jaccard similarities of Super Modules in $M$. Second, each gene was assigned a binary membership vector of Super Module Groups in terms of its occurrences in each Super Module Group (the number of Association Modules containing the gene and belonging to the Super Module Group). Third, we retrieved unique Super Module Group membership vectors and and sorted them by the numbers of their constituting genes. The genes assigned to the top-ranking Super Module Group membership vectors form the Gene Groups.

## Summary of IHAS from TCGA

The inference outcomes of all IHAS subunits from TCGA comprise a large and complex dataset which are difficult to interpret. We have placed the complete IHAS inference outcomes in Webpage documents, put them in a repository Synapse, and created an URL https://www.stat.sinica.edu.tw/IHAS/ for public access. We refer to these Webpage documents Supplementary Data. Supplementary Data contains text files for IHAS subunits (e.g., Association Models, Association Modules, Super Modules), their visualization figures, and tables on Web pages allowing simple search of Super Modules by gene names. Below we summarize these inference outcomes.

There are totally 508021 Association Models. The numbers of effectors in Association Models follow a power-law distribution when combining all cancer types together, yet the distributions vary considerably across cancer types (S3 Fig). In some cancer types (such as ACC), the majority of Association Models possess single effectors, while in other cancer types (such as BRCA), Association Models have highly varying numbers of effectors. The Association Models are grouped into 24638 Association Modules. We report the summary statistics of the Association Modules in S1A Table.

Association Modules and samples are grouped to 217 Super Modules and 228 Sample Groups. We report the summary information of Super Modules (S1B Table), Sample Groups (S1C Table), their effectors and regulators (S1D Table), and the membership occurrence matrix of target genes (S1E Table). Elaborating putative mechanisms and functional implications underlying the associations of all Super Modules is beyond the scope of this work. To demonstrate that Super Modules capture important driver alterations and functional processes of cancer, we solicit four Super Modules and illustrate their selected effectors and target genes in S4 Fig. We summarize the function of one breast cancer Super Module below and describe other three in S1 Text (Section 2.3).

Breast cancer (BRCA) Super Module 5 (S4A Fig) consists of the following prominent effectors. MYC (regulator of chr8q CNV positive association +), TP53 (+ mutation), PIK3CA and CDH1 (- mutation) are well-known driver genes [59]. MNDA and MAGEB4 (- methylation) are myeloid cell differentiation antigen involved in chronic lymphocytic leukemia and other cancers [60] and cancer-testis antigens associated with immunotherapy treatment responses and undergoing aberrant methylation [61,62]. Mir-10a and let-7 are closely involved in various cancer-related processes (e.g., [63,64]). The target genes are highly enriched with cell cycle process.

Fig 3A displays the membership occurrence matrix $M$ of 217 Super Modules (columns) and 16860 genes (rows). There are 17 Super Module Groups and 18 Gene Groups (S1E Table). A rectangular patch marked by yellow boundaries denotes that a Gene Group is over-represented in a Super Module Group. These over-representation relations reduce $M$ to a coarse-grained 18×17 over-representation matrix (S1F Table). We also mark the cancer types of sorted Super Modules and find no obvious dependency between cancer types and Super Module Groups.

## Functional characterization of IHAS

We gave biological interpretations of IHAS with three characteristics: functional enrichment of Gene Groups and Super Modules, recurrent effectors of Super Module Groups, and an Artery Network spanned by explanatory paths of associations. In addition, we justified the benefits of the hierarchical structure by showing the information gain when traversing upward and downward along the hierarchy.

**Gene Groups and Super Modules are enriched with functional categories universal and specific to cancer types.** We calculated False Discovery Rate (FDR) adjusted hyper-

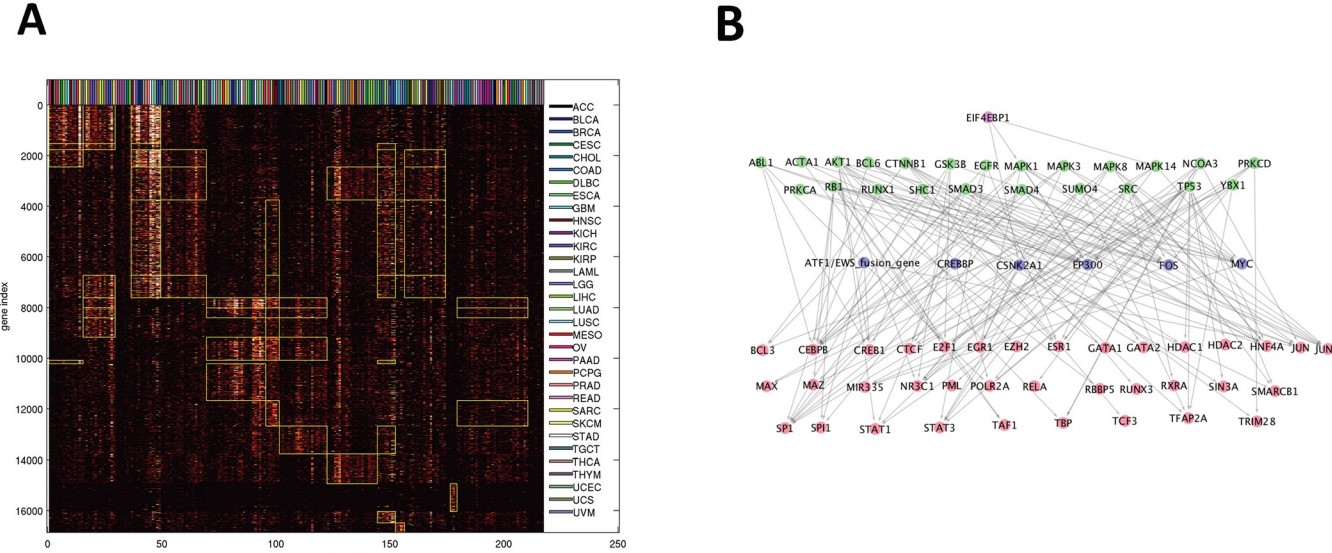

**Fig 3. Summary of integrated hierarchical association structure. A**: Memberships of 16860 target genes in 217 Super Modules. Super Modules (columns) are sorted by Super Module Groups, and genes (rows) are sorted by Gene Groups. The heat color of an entry reflects the number of modules encompassing a gene and belonging to a Super Module. A rectangular patch marked by yellow boundaries denotes that a Gene Group is over-represented in a Super Module Group. The cancer types of Super Modules are displayed on the top. **B**: Consensus Artery Network comprising indispensable interactions for explaining (effector, target) association pairs across multiple cancer types. The subnetwork spanned by highly connected hubs is displayed. Node colors denote hub levels: 1 (red), 2 (purple), 3 (green), 4 (magenta).

geometric enrichment p-values of 14545 Gene Sets from the MSigDB database [65] in Gene Groups (S2A Table). The long list of enriched Gene Sets is summarized as several representative functional classes in Table 3. Gene Groups 1–3 are highly enriched with immune responses and cell adhesion. Gene Groups 4–6 are highly enriched with cell adhesion, neurogenesis, and development. Gene Groups 7, 8, 10 and 12 are highly enriched with cell cycle process. We further aggregated them into Meta Gene Groups 1–3 respectively. Individual Gene Groups are also enriched with specific functional categories, including inflammatory responses (Gene Group 1), interferon signaling (2), generic transcription (4), synapse assembly (5), cell junction (6), chromosome segregation and cell division (7), translation and ribosomes (9), DNA repair and RNA splicing (10 and 12), respiration (14), and olfactory receptors (16).

We also calculated Gene Set enrichment p-values in Super Modules (S2B Table). Most highly enriched Gene Sets of Super Modules are compatible with the representative functional classes of the Gene Groups they possess. However, some Super Modules possess unique functional enrichment irreducible to that of Gene Groups (S2C Table). For instance, BRCA Super Modules 1–2 (Super Modules 12–13) are enriched with estrogen response genes, and LIHC Super Module 1 (Super Module 106) is enriched with lipid/fatty acid metabolism genes. These uniquely enriched functions are mostly related to the tissues of origins of specific cancer types.

**Recurrent Effectors of Super Module Groups hit the pathways of Jak-Stat, TGF-β, PI3K-AKT, MAP/ERK, apoptosis, and others.** We identified *Recurrent Effectors* of each Super Module Group whose occurrence frequencies were statistically significant. S3 Table reports the complete list of Recurrent Effectors of all Super Module Groups, and Table 4 shows the top-ranking Recurrent Effectors. Below we describe some prominent Recurrent Effectors.

In Super Module Group 1, chr9p (1-38Mb) and chr4q (90-190Mb) CNV positive associations occur in 9 and 6 Super Modules respectively. Their deletions likely down-regulate target genes via putative regulators JAK2 (chr9p) in the Jak-Stat pathway and SMARCA5, SMAD1

**Table 3. The $-\log_{10}$(enrichment p-values) of selected Gene Sets for the 18 Gene Groups.**

| gene set | g1 | g2 | g3 | g4 | g5 | g6 | g7 | g8 | g9 | g10 | g11 | g12 | g13 | g14 | g15 | g16 | g17 | g18 |
|---|---|---|---|---|---|---|---|---|---|---|---|---|---|---|---|---|---|---|
| GO-IMMUNE-RESPONSE | 183.65 | 23.53 | 14.78 | 0 | 0 | 0.28 | 0 | 0 | 0 | 0 | 3.98 | 0 | 0 | 0 | 0 | 0 | 0 | 0 |
| GO-CELL-PART-MORPHOGENESIS | 0.85 | 0 | 1.65 | 7.23 | 18.91 | 7.24 | 0 | 0.02 | 0 | 0 | 0 | 0 | 0.17 | 0 | 2.00 | 0 | 1.06 | 0.33 |
| GO-CELL-PROJECTION | 6.80 | 0.38 | 7.95 | 11.56 | 33.18 | 19.85 | 0 | 0.88 | 0.79 | 0 | 0 | 0 | 0.08 | 0 | 0.24 | 0 | 3.21 | 5.46 |
| GO-CELL-CYCLE | 0 | 0 | 0 | 3.05 | 0 | 0.94 | 70.77 | 12.46 | 0.81 | 37.70 | 0 | 31.81 | 0 | 0.47 | 7.47 | 0 | 0 | 0 |
| HALLMARK-E2F-TARGETS | 0 | 0 | 0 | 0 | 0 | 0 | 60.03 | 6.04 | 0 | 26.20 | 0 | 18.40 | 0 | 0 | 0 | 0 | 0 | 0 |
| HALLMARK-MYC-TARGETS-V1 | 0 | 0 | 0 | 0 | 0 | 0 | 14.50 | 6.04 | 1.03 | 14.51 | 0 | 57.02 | 0 | 0 | 0 | 0 | 0 | 0 |
| GO-INFLAMMATORY-RESPONSE | 94.28 | 3.70 | 7.30 | 0 | 0.10 | 0.87 | 0 | 0 | 0 | 0 | 0.23 | 0 | 0 | 0 | 0 | 0 | 0 | 0 |
| GO-CELL-CELL-ADHESION | 54.42 | 2.93 | 10.42 | 96 | 11.64 | 3.65 | 0 | 0.04 | 0 | 0 | 0 | 0 | 0 | 0 | 0 | 0 | 0 | 0.03 |
| REACTOME-INTERFERON-ALPHA-BETA-SIGNALING | 4.12 | 14.80 | 1.72 | 0 | 0 | 0 | 0 | 0 | 0.03 | 0 | 0.66 | 0 | 0 | 0 | 0 | 2.64 | 0 | 0 |
| RESPONSE-TO-TYPE-I-INTERFERON | 3.70 | 18.56 | 2.72 | 0 | 0 | 0 | 0 | 0 | 0.01 | 0 | 1.27 | 0 | 0 | 0 | 0 | 2.30 | 0 | 0 |
| REACTOME-GENERIC-TRANSCRIPTION-PATHWAY | 0 | 0 | 0 | 29.37 | 0.43 | 0.05 | 0 | 0.25 | 0 | 1.83 | 0 | 0.81 | 5.00 | 0 | 7.20 | 0 | 0 | 0 |
| GO-NUCLEIC-ACID-BINDING | 1.01 | 0.25 | 2.77 | 26.15 | 4.81 | 2.18 | 0.05 | 0.26 | 0.02 | 2.17 | 0 | 0.06 | 3.21 | 0 | 3.31 | 0 | 0 | 0 |
| GO-NEURON-DEVELOPMENT | 3.51 | 0 | 2.18 | 4.45 | 24.19 | 9.98 | 0 | 0 | 0.09 | 0 | 0 | 0 | 0 | 0 | 0.26 | 0 | 0 | 5.45 |
| GO-SYNAPSE-ASSEMBLY | 0.01 | 0 | 0 | 2.03 | 14.55 | 0.43 | 0 | 0.17 | 0 | 0 | 0 | 0 | 0 | 0 | 0 | 0 | 0 | 0.15 |
| GO-ACTIN-CYTOSKELETON | 6.87 | 2.23 | 2.49 | 4.64 | 6.38 | 10.18 | 0.36 | 0.30 | 0.19 | 0 | 0 | 0.31 | 0 | 0 | 0 | 0 | 0 | 0 |
| GO-CELL-JUNCTION | 16.52 | 1.73 | 9.09 | 4.27 | 15.97 | 25.37 | 0 | 1.97 | 4.05 | 0 | 0 | 0 | 0 | 0 | 0.07 | 0 | 0 | 5.69 |
| GO-CHROMOSOME-SEGREGATION | 0 | 0 | 0 | 0 | 0 | 0 | 43.49 | 2.77 | 0 | 9.32 | 0 | 9.39 | 0 | 0 | 2.55 | 0 | 0 | 0 |
| GO-CELL-DIVISION | 0 | 0 | 0 | 1.66 | 0 | 3.25 | 40.91 | 4.85 | 0 | 11.61 | 0 | 13.53 | 0.10 | 0.29 | 0.96 | 0 | 0.08 | 0 |
| GO-TRANSLATIONAL-INITIATION | 0 | 0 | 0 | 0 | 0 | 0 | 0.93 | 3.18 | 21.26 | 0 | 0 | 3.01 | 0 | 16.95 | 0 | 0 | 0 | 0 |
| GO-CYTOSOLIC-RIBOSOME | 0 | 0 | 0 | 0 | 0 | 0 | 0.46 | 1.46 | 25.62 | 0 | 0 | 0.47 | 0 | 17.14 | 0 | 0 | 0 | 0 |
| GO-DNA-REPAIR | 0 | 0 | 0 | 0.02 | 0 | 0 | 14.18 | 2.00 | 0.97 | 29.03 | 0 | 21.08 | 0.20 | 3.35 | 5.49 | 0 | 0 | 0 |
| GO-DNA-RECOMBINATION | 0 | 0 | 0 | 0 | 0 | 0 | 9.42 | 0.13 | 0.21 | 23.44 | 0 | 9.45 | 0 | 0 | 0.55 | 0 | 0 | 0 |
| GO-MRNA-PROCESSING | 0 | 0 | 0 | 0 | 0 | 0 | 2.84 | 1.33 | 0.24 | 21.18 | 0 | 41.93 | 0 | 8.36 | 3.02 | 0 | 0 | 0 |
| GO-RNA-SPLICING | 0 | 0 | 0 | 0 | 0 | 0 | 3.11 | 1.43 | 0.34 | 21.09 | 0 | 43.83 | 0 | 7.50 | 0.91 | 0 | 0 | 0 |
| GO-RESPIRATORY-CHAIN | 0 | 0 | 0 | 0 | 0 | 0 | 0 | 0 | 0 | 0.09 | 0 | 4.82 | 0 | 34.11 | 0 | 0 | 0.17 | 0 |
| GO-ELECTRON-TRANSPORT-CHAIN | 0 | 0 | 0.05 | 0 | 0 | 0 | 0 | 0 | 0.05 | 0.18 | 0 | 5.71 | 0 | 34.14 | 0 | 0 | 0 | 0 |
| KEGG-OLFACTORY-TRANSDUCTION | 0 | 0 | 0 | 0 | 0 | 0 | 0 | 0 | 0 | 0 | 0 | 0 | 0 | 0 | 0 | 172.05 | 0 | 0 |

**Table 4. The top-ranking Recurrent Effectors for the 17 Super Module Groups.** Numbers within parentheses indicate the occurrences in Super Modules within a Super Module Group.

| group | trans CNV | mutation | methylation | mir | phosphorylation |
|---|---|---|---|---|---|
| 1 | chr9 1Mb-21Mb (9) | TTN—(4) | CCRL2 (4) | MIR96 (5) | ESR1-S118 - (5) |
| | chr9 27Mb-38Mb (9) | SYNE1 - (3) | ICAM4 (4) | MIR93 (4) | RPS6-S235S236 + (5) |
| | chr9 21Mb-22Mb (9) | | | MIR32 (4) | CDKN1B-T198 + (5) |
| | chr9 23Mb-27Mb (7) | | | | NFKB1-S536 + (4) |
| | chr9 38Mb-141Mb (7) | | | | RPS6-S240S244 + (4) |
| | chr4 85Mb-90Mb (7) | | | | MAPK3-T202Y204 + (4) |
| | chr8 8Mb-12Mb (7) | | | | MAPK1-T202Y204 + (4) |
| | chr4 71Mb-85Mb (6) | | | | MYH9-S1943 + (4) |
| | chr4 90Mb-111Mb (6) | | | | |
| | chr4 130Mb-151Mb (5) | | | | |
| | chr4 173Mb-190Mb (5) | | | | |
| | chr5 54Mb-68Mb (5) | | | | |
| | chr5 131Mb-142Mb (5) | | | | |
| | chr8 1Mb-2Mb (5) | | | | |
| | chr8 12Mb-36Mb (5) | | | | |
| | chr17 1Mb-22Mb (5) | | | | |
| | chr22 24Mb-51Mb (5) | | | | |
| | chr1 2Mb-6Mb (5) | | | | |
| | chr4 1Mb-10Mb (5) | | | | |
| | chr4 13Mb-49Mb (5) | | | | |
| 2 | chr6 66Mb-78Mb (4) | TTN + (3) | ADAM28 (4) | | YAP1-S127 + (4) |
| | chr6 79Mb-83Mb (4) | | IL12RB1 (4) | | |
| | chr7 55Mb-57Mb (4) | | | | |
| | chr7 65Mb-159Mb (4) | | | | |
| | chr12 3Mb-8Mb (4) | | | | |
| | chr12 9Mb-11Mb (4) | | | | |
| | chr12 59Mb-70Mb (4) | | | | |
| | chr17 1Mb-7Mb (4) | | | | |
| 3 | | | | | |
| 4 | chr13 33Mb-38Mb (8) | SCN5A - (4) | HIST1H4F (7) | MIR429 (10) | ACACA-S79 - (8) |
| | chr8 1Mb-2Mb (7) | APBA2 - (4) | OTX2 (7) | MIR96 (10) | ACACB-S79 - (7) |
| | chr8 8Mb-12Mb (7) | CHD5 - (3) | PAX3 (6) | MIR425 (9) | RB1-S807S811 - (7) |
| | chr13 23Mb-33Mb (7) | FLG—(3) | TFAP2B (6) | MIR7-1 (9) | STAT3-Y705 + (7) |
| | chr13 38Mb-48Mb (7) | PIK3CA—(3) | HOXA9 (6) | MIR1307 (9) | PRKCA-S657 + (7) |
| | chr13 49Mb-53Mb (7) | CSMD1 - (3) | SYCP1 (5) | MIR629 (9) | FOXO3-S318S321 + (5) |
| | chr5 52Mb-130Mb (6) | PKD1 - (3) | HOXD10 (5) | MIR33A (9) | ESR1-S118 - (5) |
| | chr5 131Mb-142Mb (6) | GTF3C1 - (3) | ZIC1 (5) | MIR590 (8) | MAPK3-T202Y204 + (5) |
| | chr5 154Mb-180Mb (6) | TP53 - (3) | GHSR (5) | MIR25 (8) | MAPK1-T202Y204 + (5) |
| | chr8 2Mb-8Mb (6) | | POU4F2 (5) | MIR93 (8) | PRKCD-S664 + (4) |
| | chr8 12Mb-38Mb (6) | | SPARC (5) | MIR335 (8) | MAPK14-T180Y182 + (4) |
| | chr9 22Mb-23Mb (6) | | GCM2 (5) | MIR671 (8) | CDK1-Y15 - (4) |
| | chr13 95Mb-115Mb (6) | | OLIG3 (5) | MIR19A (8) | YAP1-S127 + (4) |
| | chr18 2Mb-15Mb (6) | | TLX1 (5) | MIR17 (8) | AKT1-T308 + (4) |
| | chr18 19Mb-21Mb (6) | | FOXG1 (5) | MIR20A (8) | SRC-Y527 + (4) |
| | chr9 1Mb-22Mb (6) | | MFAP4 (5) | MIR33B (8) | |
| | chr9 23Mb-141Mb (6) | | CHAD (5) | MIR454 (8) | |

*(Continued)*

**Table 4.** (Continued)

| group | trans CNV | mutation | methylation | mir | phosphorylation |
|---|---|---|---|---|---|
| | chr13 48Mb-49Mb (6) | | C1orf114 (4) | MIR15B (7) | |
| | chr16 47Mb-90Mb (6) | | TNFSF4 (4) | MIR576 (7) | |
| | chr18 15Mb-19Mb (6) | | SULT1C4 (4) | MIR32 (7) | |
| 5 | | NFE2L2 - (5) | | MIR942 (7) | BAD-S112 + (7) |
| | | KRAS—(5) | | MIR20A (7) | NFKB1-S536 + (6) |
| | | KMT2D - (5) | | MIR31 (6) | YAP1-S127 + (6) |
| | | HTT—(4) | | MIR7-1 (6) | TSC2-T1462 + (6) |
| | | PLEC—(4) | | MIR17 (6) | PDK1-S241 + (5) |
| | | NOTCH1 - (4) | | MIR625 (6) | PRKAA1-T172 + (5) |
| | | SIN3A - (4) | | MIR21 (6) | GSK3B-S9S21 + (4) |
| | | BRD4 - (4) | | MIR425 (5) | GSK3B-S9 + (4) |
| | | PTPN14 - (3) | | MIR589 (5) | FOXO3-S318S321 + (4) |
| | | COL6A3 - (3) | | MIR590 (5) | EGFR-Y1068 + (4) |
| | | CTBP1 - (3) | | MIR32 (5) | RPS6-S235S236 - (4) |
| | | PTPN13 - (3) | | MIR19A (5) | AKT1-T308 + (4) |
| | | SPEF2 - (3) | | MIR1976 (4) | PRKCA-S657 + (4) |
| | | APC—(3) | | MIR181A1 (4) | AKT2-T308 + (4) |
| | | TFAP2B - (3) | | MIR15B (4) | GSK3A-S9 + (4) |
| | | DST—(3) | | MIR576 (4) | AKT1S1-T246 + (4) |
| | | GRIK2 - (3) | | MIR550A1 (4) | CHEK2-T68 - (4) |
| | | ESR1 - (3) | | MIR550A2 (4) | MYH9-S1943 - (4) |
| | | IKZF1 - (3) | | MIR671 (4) | |
| | | EGFR—(3) | | MIR181A2 (4) | |
| group | trans CNV | mutation | methylation | mir | phosphorylation |
| 6 | chr8 41Mb-43Mb (14) | TP53 + (10) | CD1C (8) | MIR497 (7) | FOXO3-S318S321 - (10) |
| | chr12 3Mb-8Mb (11) | EPHB1 + (5) | MNDA (8) | MIR664A (6) | PRKCD-S664 - (7) |
| | chr20 1Mb-62Mb (11) | NOTCH1 + (5) | CD1B (7) | MIR383 (6) | RICTOR-T1135 - (7) |
| | chr1 146Mb-149Mb (10) | ABCC9 + (5) | SPTA1 (6) | MIRLET7B (6) | YAP1-S127 - (7) |
| | chr12 8Mb-33Mb (10) | BRCA1 + (5) | LOC730811 (6) | MIR1976 (5) | TSC2-T1462 - (6) |
| | chr1 149Mb-157Mb (10) | CHD5 + (4) | REG3A (6) | MIR490 (5) | PRKCB-S660 - (6) |
| | chr8 138Mb-146Mb (10) | MACF1 + (4) | LOC286094 (6) | MIR326 (5) | PRKCA-S657 - (6) |
| | chr19 29Mb-51Mb (10) | PTPN13 + (4) | C13orf28 (6) | MIR1247 (5) | MYH9-S1943 + (6) |
| | chr12 55Mb-56Mb (10) | DSP + (4) | GP2 (6) | MIR362 (5) | ARAF-S299 - (6) |
| | chr19 2Mb-22Mb (10) | SYNE1 + (4) | MIR770 (6) | MIR1468 (5) | RAF1-S338 - (5) |
| | chr1 157Mb-186Mb (9) | KMT2C + (4) | FCRL3 (5) | MIR1258 (4) | MAPK14-T180Y182 - (5) |
| | chr1 192Mb-215Mb (9) | ANK1 + (4) | FCER1A (5) | MIR23B (4) | EGFR-Y1068 - (5) |
| | chr1 216Mb-249Mb (9) | KMT2D + (4) | APCS (5) | MIR1287 (4) | EIF4EBP1-S65 + (5) |
| | chr6 45Mb-58Mb (9) | MYH7 + (4) | OR2T6 (5) | MIR10A (4) | BAD-S112 - (5) |
| | chr7 55Mb-57Mb (8) | RYR3 + (4) | SNORA80B (5) | MIR501 (4) | RB1-S807S811 + (5) |
| | chr8 46Mb-138Mb (8) | SMAD4 + (4) | PAX4 (5) | | AKT1-S473 - (5) |
| | chr12 38Mb-47Mb (7) | MED12 + (4) | PRSS1 (5) | | CHEK2-T68 + (5) |
| | chr12 48Mb-55Mb (7) | HFM1 + (3) | MS4A3 (5) | | MTOR-S2448 - (4) |
| | chr12 56Mb-58Mb (7) | SPTA1 + (3) | CCL11 (5) | | RPS6KA1-T359S363 - (4) |
| | chr12 129Mb-133Mb (7) | IFI16 + (3) | C20orf71 (5) | | EIF4EBP1-T70 + (4) |
| 7 | chr1 146Mb-149Mb (4) | FLG + (3) | | | |
| | | PLXNA2 + (3) | | | |
| | | OBSCN + (3) | | | |

(*Continued*)

**Table 4.** (Continued)

| group | trans CNV | mutation | methylation | mir | phosphorylation |
|---|---|---|---|---|---|
| | | ROS1 + (3) | | | |
| | | MUC17 + (3) | | | |
| | | PTPRB + (3) | | | |
| 8 | chr8 91Mb-102Mb (4) | DCC + (4) | | MIR23B (6) | SRC-Y416 - (9) |
| | chr8 104Mb-146Mb (4) | SPEN + (3) | | MIR664A (5) | STAT3-Y705 - (7) |
| | | PIK3CA + (3) | | MIR574 (5) | SHC1-Y317 - (6) |
| | | DNAH5 + (3) | | MIR1247 (5) | NFKB1-S536 - (6) |
| | | LAMA4 + (3) | | MIRLET7B (5) | SRC-Y527 - (6) |
| | | SYNE1 + (3) | | MIR653 (4) | ACACB-S79 + (5) |
| | | SVEP1 + (3) | | MIR326 (4) | ACACA-S79 + (5) |
| | | LAMA1 + (3) | | | MTOR-S2448 - (4) |
| | | PCNT + (3) | | | RPS6KA1-T359S363 - (4) |
| | | TAF1 + (3) | | | PRKAA1-T172 - (4) |
| | | | | | EGFR-Y1068 - (4) |
| | | | | | EIF4EBP1-T70 + (4) |
| | | | | | AKT1-S473 - (4) |
| | | | | | AKT1-T308 - (4) |
| | | | | | MAPK3-T202Y204 - (4) |
| | | | | | RPS6KB1-T389 - (4) |
| | | | | | PRKCA-S657 - (4) |
| | | | | | MAPK1-T202Y204 - (4) |
| | chr1 107Mb-119Mb (7) | CUBN + (4) | KIAA0408 (4) | MIR181A1 (6) | SRC-Y416 + (7) |
| | chr1 6Mb-35Mb (6) | NRAS + (3) | | MIR589 (5) | SRC-Y527 + (6) |
| | chr1 37Mb-107Mb (6) | FAT3 + (3) | | MIR590 (5) | SHC1-Y317 + (5) |
| | chr4 90Mb-111Mb (6) | KRAS—(3) | | MIR93 (5) | PRKAA1-T172 + (5) |
| | chr4 13Mb-47Mb (6) | KMT2D + (3) | | MIR17 (5) | GSK3B-S9 + (4) |
| | chr18 27Mb-35Mb (6) | CREBBP + (3) | | MIR744 (5) | NFKB1-S536 + (4) |
| | chr18 36Mb-64Mb (5) | | | MIR423 (5) | EIF4EBP1-T70 - (4) |
| | chr1 155Mb-186Mb (5) | | | MIR1976 (4) | CHEK1-S345 + (4) |
| | chr4 50Mb-58Mb (5) | | | MIR1226 (4) | CDKN1B-T198 - (4) |
| | chr4 59Mb-70Mb (5) | | | MIR28 (4) | AKT1-S473 + (4) |
| | chr4 71Mb-85Mb (5) | | | MIR574 (4) | TSC2-T1462 + (4) |
| | chr13 33Mb-38Mb (4) | | | MIR576 (4) | GSK3A-S9 + (4) |
| | chr18 2Mb-15Mb (4) | | | MIR25 (4) | |
| | chr18 21Mb-25Mb (4) | | | MIR181A2 (4) | |
| | chr18 64Mb-78Mb (4) | | | MIR511 (4) | |
| | chrX 3Mb-56Mb (4) | | | MIR326 (4) | |
| | | | | MIR19A (4) | |
| | | | | MIR625 (4) | |
| | | | | MIR655 (4) | |
| | | | | MIR21 (4) | |
| 10 | | PRKDC—(3) | CLDN4 (5) | MIR1293 (4) | ESR1-S118 + (4) |
| | | POLR2A - (3) | C4BPB (4) | | NDRG1-T346 - (4) |
| | | TP53 - (3) | LOC202781 (4) | | |
| | | | LOC145837 (4) | | |
| 11 | | | | | |
| group | trans CNV | mutation | methylation | mir | phosphorylation |

(*Continued*)

**Table 4.** (Continued)

| group | trans CNV | mutation | methylation | mir | phosphorylation |
|---|---|---|---|---|---|
| 12 | chr3 12Mb-13Mb (4) | TP53 - (4) | LOC642587 (4) | MIR561 (6) | EGFR-Y1068 + (7) |
|  |  | SPTA1 - (3) | tAKR (4) | MIR96 (5) | SHC1-Y317 + (6) |
|  |  | LRP1B - (3) | CARD17 (4) | MIR10A (5) | NDRG1-T346 + (4) |
|  |  | FAT1 - (3) | ABCA6 (4) | MIR21 (5) | MAPK3-T202Y204 + (4) |
|  |  | SYNE1 - (3) | MIR641 (4) | MIR429 (4) | ACACA-S79 + (4) |
|  |  | RB1 - (3) |  | MIR181A1 (4) | PRKCA-S657 - (4) |
|  |  | LAMA1 - (3) |  | MIR577 (4) | MAPK1-T202Y204 + (4) |
|  |  |  |  | MIR671 (4) |  |
|  |  |  |  | MIR181A2 (4) |  |
|  |  |  |  | MIR708 (4) |  |
|  |  |  |  | MIR625 (4) |  |
|  |  |  |  | MIR342 (4) |  |
|  |  |  |  | MIR1180 (4) |  |
|  |  |  |  | MIR769 (4) |  |
|  |  |  |  | MIR1468 (4) |  |
|  |  |  |  | MIR766 (4) |  |
| 13 |  |  |  |  |  |
| 14 |  |  |  |  |  |
| 15 | chr8 104Mb-146Mb (8) | NF1 - (5) |  | MIR574 (5) | YBX1-S102 + (6) |
|  | chr5 1Mb-12Mb (8) | LRP2 + (4) |  | MIR432 (5) | RAF1-S338 + (5) |
|  | chr5 32Mb-44Mb (7) | MACF1 - (3) |  | MIR758 (5) | EGFR-Y1173 + (5) |
|  | chr7 8Mb-57Mb (7) | ASTN1 + (3) |  | MIR502 (5) | ERBB3-Y1289 + (5) |
|  | chr8 46Mb-104Mb (7) | SOX11 + (3) |  | MIR532 (5) | RICTOR-T1135 + (4) |
|  | chr5 12Mb-32Mb (7) | MAP2 + (3) |  | MIR23B (4) | RPS6-S235S236 - (4) |
|  | chr5 137Mb-142Mb (7) | MUC5B + (3) |  | MIR431 (4) | RPS6-S240S244 - (4) |
|  | chr7 65Mb-159Mb (7) | KRAS + (3) |  | MIR654 (4) | TSC2-T1462 + (4) |
|  | chr20 1Mb-62Mb (7) | CREBBP + (3) |  | MIR369 (4) | PRKCB-S660 + (4) |
|  | chr5 131Mb-137Mb (7) | BRCA1 - (3) |  | MIR497 (4) | ARAF-S299 + (4) |
|  | chr5 154Mb-180Mb (6) | PTPRM + (3) |  | MIR501 (4) |  |
|  | chr7 63Mb-65Mb (6) | SMARCA4 + (3) |  | MIR362 (4) |  |
|  |  |  |  | MIR766 (4) |  |
| 16 |  |  |  |  |  |
| 17 |  |  |  |  |  |

and SMAD5 (chr4q) in the TGF-$\beta$ pathway respectively. Super Module Group 1 consists of target genes enriched with immune responses (Gene Groups 1–3).

In Super Module Group 4, prominent Recurrent Effectors include chr13p (23-53Mb) CNV positive association with putative regulators FLT3 [66], CDK8 [67], ELF1 [68], and RB1 [59], PIK3CA and TP53 mutation negative associations. Super Module Group 4 consists of target genes enriched with immune responses, cell adhesion, and development (Gene Groups 1–6).

In Super Module Group 6, prominent Recurrent Effectors include CNV positive associations of chr8q (46-138Mb) (via putative regulators MYC, MCM4 [69], CCNE2 [59], E2F5 [59]) and chr20 (1-62Mb) (via putative regulators E2F1 [59] and AURKA [70]), mutation positive associations of TP53, EPHB1 [71], NOTCH1 [59], ABCC9, and BRCA1. Super Module Group 6 consists of target genes enriched with cell cycle process (Gene Groups 7, 8, 10, 12).

**Artery Networks spanned by explanatory paths of associations traverse three Meta Gene Groups and several cancer-related pathways.** Associations between effectors and

targets are likely caused by cascades of gene regulatory links, which are realized by paths in the network of molecular interactions. We compiled a unified network of molecular interactions from several large-scale databases and datasets and identified the subnetwork traversed by the connecting paths of many association pairs. We term this subnetwork an *Artery Network* as it is vital for explaining many associations. The common portion of the Artery Networks across multiple cancer types is the *Consensus Artery Network*.

Molecules in the Artery Network are stratified by levels. Molecules at higher and lower levels are close to effectors and targets respectively. S5 Fig and S4A Table report the whole Consensus Artery Network, Fig 3B displays the subnetwork spanned by highly connected *hubs*, and S4B Table shows the summary information of hubs. Prominent hubs include POLR2A, STAT3, SP1 (level 1), EP300, MYC, FOS (level 2), and TP53, SMAD4, EIF4EBP1 (level 3). High-level hubs typically contain effectors associated with many target genes. Low-level hubs are typically master transcription factors that regulate many target genes but rarely harbor molecular alterations themselves. The highly enriched Gene Sets and pathways in the Consensus Artery Network (S4C Table) include the KEGG pathways involved in various cancer types, functions involved in the three Meta Gene Groups, and numerous oncogenic signaling pathways (such as MAPK, Wnt, EGF, PDGF, TCR, MET, BCR, PI3K, and KIT pathways).

**Benefits of the hierarchical structure.** IHAS provides benefits not revealed in a flat structure of associations. To justify these benefits we elaborated the information gain by traversing up or down along the hierarchy. The upward information gain is obvious, as a higher level subunit unifies multiple lower level subunits and thus contains information not covered by individual lower level subunits. For instance, a Super Module bundles the associations pertaining to multiple types of molecular aberrations, and a Super Module Group bundles the associations occurred in multiple cancer types. Conversely, the downward information gain offers details in lower level subunits but ignored in higher level subunits. S6 Fig summarizes the downward information gain at three levels. Panel A lists four Super Modules which are not enriched with the 18 Gene Groups, including those enriched with estrogen response genes in BRCA, lipid metabolism genes in CHOL and LIHC, and heme metabolism gene in LAML. Panel B displays the overall distribution of correlation coefficients between the mean target gene expression profiles of Association Modules and their encompassing Super Modules. While most Association Modules possess similar expression profiles with their encompassing Super Modules, some Association Modules behave differently from other members of the same Super Modules, indicating information loss when aggregating Association Modules to Super Modules. Panel C compares the correlation coefficient distributions of the target genes within the same Association Modules subdivided into two groups. Group 1 include the gene pairs with identical effectors (black curve, mean correlation coefficient 0.318) and group 2 include the gene pairs with non-identical effectors (though they share at least one common effector) (red curve, mean correlation coefficient 0.228). Their difference indicates that the genes with distinct Association Model topology possess distinct expression profiles even though they belong to the same Association Modules.

## Alignments of IHAS with clinical phenotypes

IHAS provides useful information in diagnosis about tumor molecular subtypes and survival times. Within each cancer type, we clustered samples into Sample Groups according to their combinatorial expression patterns over Super Modules. We found Sample Groups were aligned with the majority of clinical features in TCGA data. Sample Groups across cancer types were further clustered into eight Pan-cancer Sample Groups according to their combinatorial expressions of three Meta Gene Groups. The Pan-cancer Sample Groups were also

aligned with several pan-cancer phenotypes supplied in TCGA or external studies. We further assessed the relations of IHAS subunits with the survival times of patients within and across cancer types. Intriguingly, these relations categorize the tumors from 33 cancer types into a few groups which may require distinct treatments.

**Sample Groups are aligned with over 80% of clinical features within cancer types.** TCGA samples were assigned to Sample Groups and annotated with rich clinical features. The Sample Group labels of all TCGA samples are reported in S5A Table. We selected 92 features which had categorical and non-missing values in at least ¼ of the samples in the corresponding cancer types. To check whether Sample Groups were informative about the feature values, we quantified alignment between Sample Groups and feature values by concentration coefficients, which denote the fraction of samples possessing the dominant feature values in each Sample Group. For instance, when aligning ACC Sample Groups with the DNA methylation subtypes (S10A Fig), Sample Groups 1, 2 and 3 are dominated by CIMP low, intermediate and high phenotypes (23/34, 12/20, 13/24 samples respectively), and the concentration coefficient is $(23+12+13)/(34+20+24) = 48/78 = 0.6154$. We (1) summarized the aligned clinical features of all Sample Groups (Table 5), (2) visualized and reported the combinatorial expressions of sorted Super Modules and Sample Groups juxtaposed with clinical feature values of individual cancer types (Supplementary Data), (3) reported the concentration coefficients of 92 features (S5B Table). 76 of 92 (82.6%) features possess concentration coefficients $\geq 0.6$. In contrast, when randomly permuting samples in each cancer type 1000 times at most 26 (28.3%) features possess concentration coefficients $\geq 0.6$. We also clustered the mRNA expression data in each cancer type by the k-means algorithm with $k = 4$ and found 65 of 92 (70.7%) features possess concentration coefficients $\geq 0.6$. Hence the sample clusters based on mRNA data alone are aligned with most clinical features but are still inferior to the Sample Groups inferred from the association structure of the TCGA data.

We briefly elucidate the alignment outcomes of four cancer types (S7 Fig and S5C and S5D Table). In BRCA, Sample Groups demarcate PAM50 subtypes [72] luminal A (red), luminal B (cyan), basal-like (blue) and HER2-enriched (green) subtypes dominate Sample Groups 1–2 and 6, 5, 3 and 4 respectively. In COAD, Sample Groups are aligned with the consensus molecular subgroups (CMS, [73]) CMS1 (hyper mutated, blue), CMS2 (canonical, green), CMS3 (metabolic, red) and CMS4 (mesenchymal, cyan) subtypes dominate Sample Groups 2, 6–7, 3–4, 1 and 8 respectively. In LGG, Sample Groups are aligned with COC (clusters-of-clusters) subtypes and IDH mutation states. In SARC, Sample Groups are aligned with histology, PARADIGM and icluster labels. The combinatorial expression patterns of Super Modules (S5C Table) are compatible with the annotated subtypes. Feature values of individual samples are reported in S5D Table. For instance, in BRCA Sample Groups dominated by luminal A and B subtypes have positive ER states and high expressions in estrogen response enriched Super Modules (1–3). In COAD, Sample Groups dominated by CMS1 (hyper-mutated) have high expressions in cell cycle enriched Super Modules (8–10) and immune response and development enriched Super Modules (6–7). In SARC, Sample Groups dominated by dedifferentiated and undifferentiated tumors have moderate-to-high expressions in immune response and development enriched Super Modules (7–8).

**Pan-cancer Sample Groups are aligned with pan-cancer phenotypes.** We sorted and partitioned the 228 Sample Groups into 8 Pan-cancer Sample Groups in terms of the binary expression states of the three Meta Gene Groups. S5E Table reports the Pan-cancer Sample Group assignments of 228 Sample Groups, and Fig 4A displays the average expression levels of the sorted Sample Groups in the 18 Gene Groups. Pan-cancer Sample Groups 1 (001, blue, high expressions in Meta Gene Group 3 and low expressions in Meta Gene Groups 1–2) and 6 (110, yellow, high expressions in Meta Gene Groups 1–2 and low expressions in Meta Gene

**Table 5. Alignments of Sample Groups with clinical features within cancer types.** For each cancer type, the representative values (features) of selected subtypes of each Sample Group are reported.

| cancer type | sample group | subtype | subtype | subtype | subtype | subtype | subtype | subtype |
|---|---|---|---|---|---|---|---|---|
| ACC | | C1A/C1B | mRNA | methylation | SCNA | COC | | |
| | 1 | C1B | steroid-low | CIMP-low | chromosomal | COC1 | | |
| | 2 | C1B | | CIMP-med | chromosomal | COC1 | | |
| | 3 | C1A | steroid-high | CIMP-med | chromosomal | COC2 | | |
| | 4 | C1A | proliferation | CIMP-high | noisy | COC3 | | |
| BLCA | | mRNA | | | | | | |
| | 1 | 2 | | | | | | |
| | 2 | 1 | | | | | | |
| | 3 | 1 | | | | | | |
| | 4 | 1 | | | | | | |
| | 5 | 2 | | | | | | |
| | 6 | 3 | | | | | | |
| | 7 | | | | | | | |
| BRCA | | ER | PR | HER2 | PAM50 | | | |
| | 1 | + | + | - | luminal A | | | |
| | 2 | + | + | - | luminal A | | | |
| | 3 | - | - | - | basal-like | | | |
| | 4 | mixed | mixed | + | HER2-enriched | | | |
| | 5 | + | + | - | luminal A,B | | | |
| | 6 | + | + | - | luminal A,B | | | |
| CESC | | Dx | SCNA | icluster | paradigm | | | |
| | 1 | adenocarcinoma | low | C3 | C2 | | | |
| | 2 | adenocarcinoma | low | C3 | C2 | | | |
| | 3 | adenocarcinoma | low | C3 | C2 | | | |
| | 4 | squamous | high | C2 | C1 | | | |
| | 5 | squamous | high | C2 | C3 | | | |
| | 6 | squamous | high | C2 | C4 | | | |
| | 7 | squamous | high | C2 | | | | |
| | 8 | squamous | mixed | C2 | C1 | | | |
| COAD | | MSI | MLH1-silencing | mRNA | hypermutated | CIMP | CMS | |
| | 1 | MSI-L | 0 | MSI/CIMP | mixed | CIMP.Neg | CMS4 | |
| | 2 | MSI-H | 1 | MSI/CIMP | | 1 CIMP.High | CMS1 | |
| | 3 | | | | | | CMS3 | |
| | 4 | mixed | 0 | invasive | | 0 CIMP.Neg | CMS3 | |
| | 5 | MSS | 0 | invasive | | 0 CIMP.Neg | mixed | |
| | 6 | MSS | 0 | CIN | | 0 CIMP.Neg | CMS2 | |
| | 7 | MSS | 0 | CIN | | 0 CIMP.Neg | CMS2 | |
| | 8 | MSS | 0 | mixed | | 0 CIMP.Neg | CMS4 | |
| | 9 | MSS | 0 | mixed | | 0 CIMP.Neg | mixed | |
| ESCA | | histology | ESCC subtype | methylation | SCNA | mRNA | mir | |
| | 1 | ESCC | C1,C2 | C2 | C3 | C2,C3 | C3 | |
| | 2 | ESCC | C1 | C2 | C3 | C3 | C2 | |
| | 3 | ESCC | C1 | C2 | C3 | C3 | C3 | |
| | 4 | EAC | | C1 | C1,C2 | C1 | C1 | |
| | 5 | EAC | | C1 | C1,C2 | C1 | C1 | |
| | 6 | EAC | | C1 | C1,C2 | C1 | C1 | |
| | 7 | EAC | | C1 | C1,C2 | C1 | C1 | |
| GBM | | CIMP | mRNA | | | | | |
| | 1 | | | | | | | |
| | 2 | G-CIMP | G-CIMP | | | | | |
| | 3 | G-CIMP | G-CIMP | | | | | |

*(Continued)*

**Table 5.** (Continued)

| cancer type | sample group | subtype | subtype | subtype | subtype | subtype | subtype | subtype |
|---|---|---|---|---|---|---|---|---|
| | 4 | non-G-CIMP | proneural | | | | | |
| | 5 | non-G-CIMP | proneural | | | | | |
| | 6 | non-G-CIMP | proneural | | | | | |
| | 7 | G-CIMP | G-CIMP | | | | | |
| | 8 | non-G-CIMP | proneural,neural | | | | | |
| | 9 | non-G-CIMP | mixed | | | | | |
| | 10 | non-G-CIMP | classical | | | | | |
| | 11 | non-G-CIMP | classical | | | | | |
| | 12 | non-G-CIMP | classical | | | | | |
| | 13 | non-G-CIMP | mesenchymal | | | | | |
| | 14 | non-G-CIMP | mesenchymal | | | | | |
| | 15 | non-G-CIMP | mesenchymal | | | | | |
| HNSC | | mRNA | paradigm | | | | | |
| | 1 | basal | C4 | | | | | |
| | 2 | basal,mesenchy. | C4 | | | | | |
| | 3 | mesenchymal | C1 | | | | | |
| | 4 | basal | C4 | | | | | |
| | 5 | | | | | | | |
| | 6 | mesenchy.,atypical | C1, C5 | | | | | |
| | 7 | mixed | mixed | | | | | |
| | 8 | classical | C2, C5 | | | | | |
| | 9 | atypical,mesenchy. | C5 | | | | | |
| cancer type | sample group | subtype | subtype | subtype | subtype | subtype | subtype | subtype |
| KIRC | | mRNA | | | | | | |
| | 1 | C1 | | | | | | |
| | 2 | | | | | | | |
| | 3 | C1 | | | | | | |
| | 4 | C1 | | | | | | |
| | 5 | C4, C1 | | | | | | |
| | 6 | C2 | | | | | | |
| | 7 | C2, C4 | | | | | | |
| | 8 | C3 | | | | | | |
| KIRP | | mRNA | RPPA | COC | | | | |
| | 1 | C2 | C3 | C2 | | | | |
| | 2 | C2 | C3 | C2 | | | | |
| | 3 | C2 | mixed | C2 | | | | |
| | 4 | C2 | mixed | C2 | | | | |
| | 5 | C1 | C2 | C1 | | | | |
| | 6 | C1 | C2 | C1 | | | | |
| | 7 | C1 | mixed | C1 | | | | |
| | 8 | mixed | C3 | C1 | | | | |
| LAML | | mRNA | | | | | | |
| | 1 | mixed | | | | | | |
| | 2 | mixed | | | | | | |
| | 3 | C5 | | | | | | |
| | 4 | C5 | | | | | | |
| | 5 | mixed | | | | | | |
| | 6 | C4 | | | | | | |
| | 7 | C6, C3 | | | | | | |
| | 8 | C6 | | | | | | |
| | 9 | C3 | | | | | | |
| | 10 | mixed | | | | | | |

(Continued)

**Table 5.** (Continued)

| cancer type | sample group | subtype | subtype | subtype | subtype | subtype | subtype | subtype |
|---|---|---|---|---|---|---|---|---|
| LGG | | IDH | mRNA | methylation | mir | SCNA | oncosign | COCcluster |
| | 1 | IDHwt | R2 | M4 | mixed | C2 | P3 | coc2 |
| | 2 | IDHwt | | M4 | mi2 | C2 | | coc2 |
| | 3 | IDHmut-noncodel | R1 | M5 | mi2 | C1 | P1 | coc1 |
| | 4 | IDHmut-noncodel | R1 | M3, M5 | mi2 | C1 | P2 | coc1 |
| | 5 | IDHmut-noncodel | R1 | M3, M5 | mi2 | C1 | P2 | coc1 |
| | 6 | IDHmut-codel | R3 | M2, M3 | mi2 | C3 | P2 | coc3 |
| | 7 | IDHmut-both | R4 | M3, M5 | mi2 | C3, C1 | P4 | coc3, coc1 |
| | 8 | mixed | R4 | mixed | mi1 | C3, C1 | P4 | coc3, coc1 |
| LIHC | | mRNA | paradigm | icluster | | | | |
| | 1 | mixed | mixed | mixed | | | | |
| | 2 | mixed | C1, C4 | mixed | | | | |
| | 3 | C3, C4 | mixed | C2, C3 | | | | |
| | 4 | C5 | C1, C4 | C2, C3 | | | | |
| | 5 | C4 | C2 | C3 | | | | |
| | 6 | mixed | C4, C2 | C1, C2 | | | | |
| | 7 | C1 | C3 | C1 | | | | |
| | 8 | C1 | C2, C3 | C1, C3 | | | | |
| | 9 | C1 | C2, C3 | C1, C3 | | | | |
| LUAD | | icluster | | | | | | |
| | 1 | C5, C6 | | | | | | |
| | 2 | C5 | | | | | | |
| | 3 | C1, C4 | | | | | | |
| | 4 | C3 | | | | | | |
| | 5 | C3, C6 | | | | | | |
| | 6 | C3, C6 | | | | | | |
| | 7 | C2 | | | | | | |
| LUSC | | subtype | | | | | | |
| | 1 | | | | | | | |
| | 2 | secretory | | | | | | |
| | 3 | basal, secretory | | | | | | |
| | 4 | basal, primitive | | | | | | |
| | 5 | classical | | | | | | |
| | 6 | classical | | | | | | |
| PAAD | | mRNA2 | mRNA3 | methylation1 | | | | |
| | 1 | | | | | | | |
| | 2 | C2 | C3 | lowpurity | | | | |
| | 3 | mixed | C2, C3 | lowpurity | | | | |
| | 4 | C2 | C3 | lowpurity | | | | |
| | 5 | C1 | C1, C3 | C2 | | | | |
| | 6 | mixed | C1 | C1, C2 | | | | |
| | 7 | C1, C2 | C3 | C2 | | | | |
| | 8 | C1 | C3 | C1 | | | | |
| PCPG | | mRNA | methylation | | | | | |
| | 1 | cortical admixture | low-methylated | | | | | |
| | 2 | pseudohypoxia | intermediate | | | | | |
| | 3 | pseudohypoxia | intermediate | | | | | |
| | 4 | pseudohypoxia | hyper-methylated | | | | | |
| | 5 | mixed | mixed | | | | | |
| | 6 | kinase signaling | low-methylated | | | | | |
| | 7 | kinase signaling | low-methylated | | | | | |

*(Continued)*

**Table 5.** (Continued)

| PRAD | | icluster | mRNA | | | | | |
|---|---|---|---|---|---|---|---|---|
| | 1 | mixed | mixed | | | | | |
| | 2 | C3 | C3 | | | | | |
| | 3 | C3 | C3 | | | | | |
| | 4 | C3 | C3 | | | | | |
| | 5 | C1 | C1 | | | | | |
| | 6 | | | | | | | |
| | 7 | C1, C2 | C1, C2 | | | | | |
| READ | | CMS | | | | | | |
| | 1 | CMS4 | | | | | | |
| | 2 | mixed | | | | | | |
| | 3 | CMS2 | | | | | | |
| | 4 | CMS2, CMS3 | | | | | | |
| | 5 | mixed | | | | | | |
| SARC | | histology | methylation | mRNA | mir | RPPA | paradigm | icluster |
| | 1 | ULMS, STLMS | M3, M5 | C3 | C1 | C1 | C3 | C1 |
| | 2 | STLMS, ULMS | M3 | C3 | C1 | C1 | C3 | C1 |
| | 3 | ULMS, SS | M5, M2 | C3, C2 | C1, C2 | mixed | C2 | C1, C2 |
| | 4 | mixed | M1 | C2, C1 | C2 | C5, C4 | C2, C4 | C4, C2, C3 |
| | 5 | DDLPS | M1 | C1 | C2 | C3, C4, C5 | C4 | C3 |
| | 6 | UPS, MFS | M4 | C1 | C2 | C2 | C5 | C5 |
| | 7 | UPS, MFS | M1, M4 | C1 | C2 | C2, C3 | C1 | C5, C3 |
| THCA | | histology | mRNA | mir | methylation | RPPA-number | | |
| | 1 | follicular | C1 | C4 | follicular,CpG | C4, C1 | | |
| | 2 | classical | C1, C5 | C5, C4 | class. 2,follicular | mixed | | |
| | 3 | follicular | C1, C5 | C4 | class. 2,follicular | C1 | | |
| | 4 | classical | C5, C2 | mixed | class. 2,follicular | mixed | | |
| | 5 | classical, tall | C4 | C2 | classical 2 | C4 | | |
| | 6 | classical | C3 | C3 | classical 2 | C3, C4 | | |
| | 7 | classical | C5 | mixed | classical 2 | mixed | | |
| | 8 | classical | C4 | C2 | classical 1,2 | C2 | | |
| UCEC | | icluster | mRNA | histology | | | | |
| | 1 | notassigned(NA) | mixex | endometrioid | | | | |
| | 2 | NA,CN low | C2 | endometrioid | | | | |
| | 3 | mixed | C1 | serous | | | | |
| | 4 | mixed | C2, C3 | endometrioid | | | | |
| | 5 | mixed | C3 | endometrioid | | | | |
| | 6 | CN high | C1 | serous | | | | |
| | 7 | NA | C1, C2 | endometrioid | | | | |
| | 8 | MSI,NA | C2 | endometrioid | | | | |
| | 9 | POLE,NA | C1 | endometrioid | | | | |
| UCS | | methylation | | | | | | |
| | 1 | mixed | | | | | | |
| | 2 | C1, C3 | | | | | | |
| | 3 | C1 | | | | | | |
| | 4 | C2 | | | | | | |
| UVM | | SCNA | methylation | lncRNA | mRNA | paradigm | | |
| | 1 | C1 | C1 | C1, C2 | C1, C2 | C1, C2 | | |
| | 2 | C2 | C2 | C2, C1 | C2, C1 | C2, C1 | | |
| | 3 | C3 | C4 | C3 | C3 | C3 | | |
| | 4 | C4 | C4 | C4 | C4 | C4 | | |

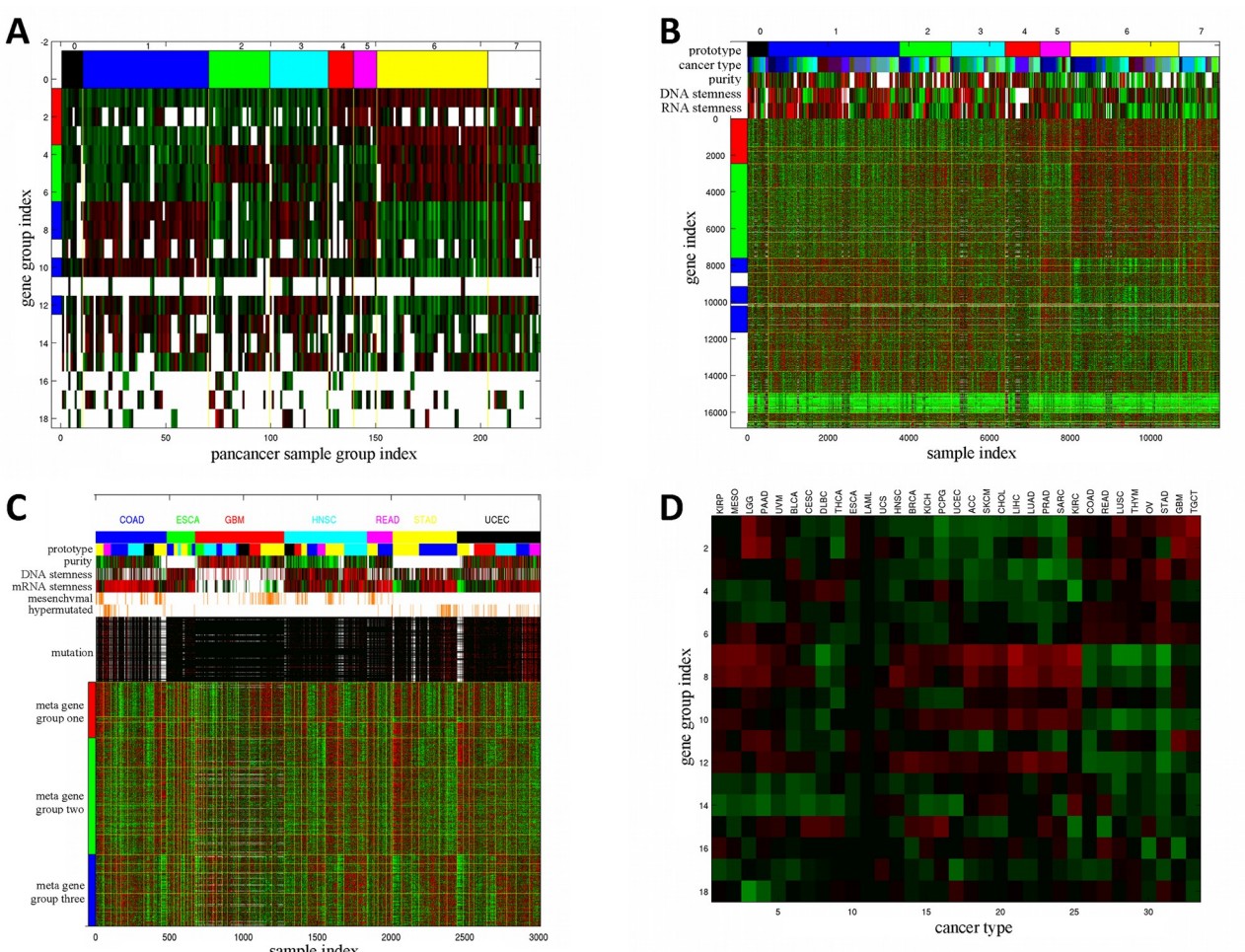

**Fig 4. Summary of Pan-cancer Sample Groups and alignments with clinical features. A**: Combinatorial expression patterns of 18 Gene Groups (rows) in 228 Sample Groups (columns). The three Meta Gene Groups are marked on the left: Meta Gene Group 1 (immune and inflammatory responses, red), Meta Gene Group 2 (embryonic development and neurogenesis, green), Meta Gene Group 3 (cell cycle process and DNA repair, blue). Sample Groups are sorted by the combinatorial expression patterns of Meta Gene Groups. The 8 Pan-cancer Sample Groups (0–7) are marked on the top and separated by yellow vertical lines. A white patch indicates that a Gene Group is not enriched in any Super Module in the cancer types of a Sample Group. **B**: Expressions of 16860 genes in 11701 samples. Genes are sorted by Gene Groups, and Meta Gene Groups are marked on the left. Samples are sorted by Sample Groups and Pan-cancer Sample Groups. Gene Groups and Pan-cancer Sample Groups are separated by yellow boundaries. The Pan-cancer Sample Groups (prototypes), cancer types, sample purity, DNA and RNA stemness of samples are displayed on the top. **C**: mRNA expressions of three Meta Gene Groups overlaid with several pan-cancer features in seven cancer types. Samples within each cancer type are sorted by Sample Groups. The following sample features are also displayed: cancer types, prototypes (Pan-cancer Sample Groups), sample purity, DNA and RNA stemness, mesenchymal states, hyper-mutated states, and mutations of selected genes. Prototype color codes follow the top row in panel **B**. **D**: Direction and strength of mRNA expression-survival time associations ($p_{diff}$ scores of Cox regression coefficient distributions) of all Gene Groups in all cancer types. Positive and negative deviations (indicating positive and negative Cox regression coefficients) have red and green colors respectively.

Group 3) comprise the most abundant Sample Groups (60 and 53 respectively). Fig 4B and S5F Table show mRNA expressions of 18 Gene Groups on 11701 TCGA samples sorted by Pan-cancer Sample Groups and several aligned phenotypes. Contrary to the integrated sample clusters derived from previous TCGA pan-cancer studies which are closely aligned with cancer types [11,12], only less than half (15 of 33) of the cancer types are significantly enriched (hyper-geometric p-values $\leq 10^{-20}$) in any of the Pan-cancer Sample Groups (S13B Table), suggesting that the combinatorial expression patterns of Meta Gene Groups are more ubiquitous across cancer types than the prior studies.'

Several molecular features are present in multiple cancer types: hyper-mutated tumors in gastrointestinal tract cancers (COAD, READ, ESCA, STAD) and uterus cancers (UCEC), mesenchymal samples in COAD, READ, GBM, and HNSC, sample purity [74] and stemness [75]. We demonstrate that the combinatorial expression patterns of Meta Gene Groups are closely aligned with these pan-cancer features. In Fig 4B, purity is strongly anti-correlated with median Meta Gene Group 1 expressions (correlation coefficient -0.5029), and RNA stemness is strongly correlated with median Meta Gene Group 3 expressions (correlation coefficient 0.5101).

Fig 4C displays mRNA expressions of three Meta Gene Groups on TCGA samples of seven cancer types sorted by Sample Groups within cancer types, Pan-cancer Sample Group (prototype) labels, and five pan-cancer phenotypes. Mesenchymal samples typically possess high expressions in Meta Gene Groups 1–2 and low expressions in Meta Gene Group 3 (Pan-cancer Sample Group/prototype 6, hyper-geometric p-value $1.80 \times 10^{-75}$). Hyper-mutated samples typically encounter frequent mutations in selected genes and possess high expressions in Meta Gene Group 3 and low expressions in Meta Gene Group 2 (Pan-cancer Sample Group/prototype 1 or 5, hyper-geometric p-values $6.07 \times 10^{-25}$ and $3.96 \times 10^{-18}$ respectively). Alignments of Meta Gene Group expressions with sample purity and stemness shown in Fig 4B are also salient in Fig 4C.

**IHAS subunits are associated with survival times.** We demonstrate that overall survival times are strongly aligned with IHAS within certain cancer types, and infer the general relations between survival times and the combinatorial expression patterns of Gene Groups.

Within each cancer type, we performed three prognostic associations. First, we estimated the direction and strength of the survival time-mRNA expression association in each Super Module by a novel measure $p_{diff}$ [76] for the deviation of Cox regression coefficients from a background distribution (S6A Table). Second, we assessed whether patients belonging to distinct Sample Groups exhibited significantly different Kaplan-Meier curves (S6A Table and Supplementary Data). Third, we manually generated a decision tree to segregate Patient Groups by their combinatorial expression patterns of Super Modules, and visualized survival analysis outcomes of six cancer types in S8 and S9 Figs and all cancer types in Supplementary Data.

We illustrate prognostic analysis outcomes in bladder cancer (S8A–S8C Fig). Super Modules 1–4 and 5–8 have strong positive and negative Cox regression coefficients respectively (S6A Table). The 5 Patient Groups possess disparate Kaplan-Meier curves (log-rank p-value $4.43 \times 10^{-7}$), and can be roughly subdivided into the lower tier (groups 3–5) and the upper tier (groups 1–2). The differences of survival times between these groups can be captured by the differences in their Super Module gene expressions and represented as a decision tree.

We also evaluated the $p_{diff}$ scores of survival time-mRNA expression associations of 18 Gene Groups (Fig 4D and S6B Table). For each Gene Group the direction of associations with survival times varies with cancer types. Meta Gene Groups 1–2 members possess similar $p_{diff}$ score variations but are anti-correlated with Meta Gene Group 3 members. In addition, cancer types are subdivided into three groups according to their variations of $p_{diff}$ scores over Gene Groups. In group 1 cancers (KIRP, MESO, LGG and PAAD), Meta Gene Groups 1 and 3 members have positive Cox regression coefficients and Meta Gene Group 2 members have negative Cox regression coefficients. In group 2 (HNSC, BRCA, KICH, PCPG, KIRC), Meta Gene Groups 1–2 members have negative Cox regression coefficients and Meta Gene Group 3 members have positive Cox regression coefficients. Group 3 cancers (COAD, READ, LUSC, THYM, TGCT) have the opposite directions of Cox regression coefficients from group 2.

**An integrated view of IHAS information.** So far we have separately depicted distinct aspects of IHAS subunits but not yet provided an integrated view of all components. Here we

present a holistic picture of IHAS within and across cancer types, and highlight implications in a few clinical traits. To save space we elucidate information of breast and colon cancers in Fig 5 and pan-cancer in Fig 6, and place information of other cancer types in S10 and S11 Figs and S1 Text Section 4.5.

**Integrated IHAS information in BRCA and COAD.** Fig 5A displays the following IHAS information in BRCA data: functional enrichment of Super Modules, occurrences of hubs in the Consensus Artery Network as effectors in each Super Module, combinatorial expressions of Super Modules and Sample Groups, the composition of PAM50 subtypes in each Sample Group, and the fractions of patients surpassing 5-year survival time in each Sample Group. To sum up, the PAM50 subtypes are determined by the combinatorial expression patterns of the estrogen response (ER) pathways (Super Modules 1–3) and three Meta Gene Groups (Super Modules 4–8). Both luminal A and luminal B tumors have moderate-to-high ER expressions; while most luminal A tumors (Sample Groups 1–2) have high expressions in Meta Gene Groups 1–2 and low expressions in Meta Gene Group 3, but most luminal B and some luminal A tumors (Sample Groups 5–6) have low expressions in Meta Gene Groups 1–3. Basal-like and Her2-enriched tumors (Sample Groups 3–4) have moderate-to-high expressions in Meta Gene Groups 1–3 and low ER expressions. Sample Groups 1–2 comprise slightly higher fractions of long-surviving patients (0.25–0.35) than Sample Groups 4–5 (around 0.2), suggesting that luminal A patients may possess slightly better prognostic outcomes than Her2-enriched and luminal B patients.

Fig 5B displays the integrated IHAS information in COAD data. The CMS subtypes are determined by the combinatorial expressions of the three Meta Gene Groups (Super Modules 1 and 6–10) and Gene Group 14 (Super Module 4). CMS1 (hyper mutated) and CMS4 (mesenchymal) tumors have high expressions in Meta Gene Groups 1–2; while CMS1 and some CMS4 tumors (Sample Group 1) have moderate expressions in Meta Gene Group 3, but some CMS4 (mesenchymal) (Sample Group 5) have low expressions in Meta Gene Group 3. CMS2 (canonical) tumors (Sample Groups 3–4) have moderate-to-high expressions in Meta Gene Group 2 and Gene Group 14 and low expressions in Meta Gene Groups 1 and 3. CMS3 (metabolic) and some other tumors (Sample Group 2) have moderate expressions in Meta Gene Group 3 and low expressions in Meta Gene Groups 1–2. Sample Groups 2 and 4 comprise slightly higher fractions of long-surviving patients (0.12–0.15) than others (below 0.08).

**Alterations on two sets of pathways impact different Meta Gene Groups.** We have shown the dominant functions of the target genes in Gene Groups and Super Module Groups and the recurrent effectors in Super Module Groups. It is natural to combine both information and check which pathways are frequently perturbed in selected Super Module Groups and their functional consequences. To this end we patched together four tables and visualized them in Fig 6A. The upper-left patch ($M_1$) shows the binary over-representation matrix of 18 Gene Groups in 17 Super Module Groups (S1E Table). The upper-right patch ($M_2$) shows the occurrence frequencies of 101 selected hub effectors in each Super Module Group. The lower-right patch ($M_3$) displays the membership matrix of the selected hub effectors in 35 known pathways. The lower-left patch ($M_4$) displays the aggregate occurrence frequencies of effector pathways in Gene Groups (which resembles $M_1 \cdot M_2 \cdot M_3$). Intriguingly, the 35 pathways are roughly divided into two groups. Group 1 primarily impacts Meta Gene Group 3 and includes pathways pertaining to cell cycle control (G1, G2, ATM, telomere, RB, E2F), apoptosis and senescence (P53 and RB). Group 2 primarily impacts Meta Gene Groups 1–2 and includes extracellular matrix control (ECM and integrin) and various upstream signaling pathways (such as PYK2, HER2, MAPK/ERK and MET). Sixteen hub effectors frequently occur in Super Module Groups and selected pathways, including CDKN2A, RB1, CDK2, TP53, and E2F1 in

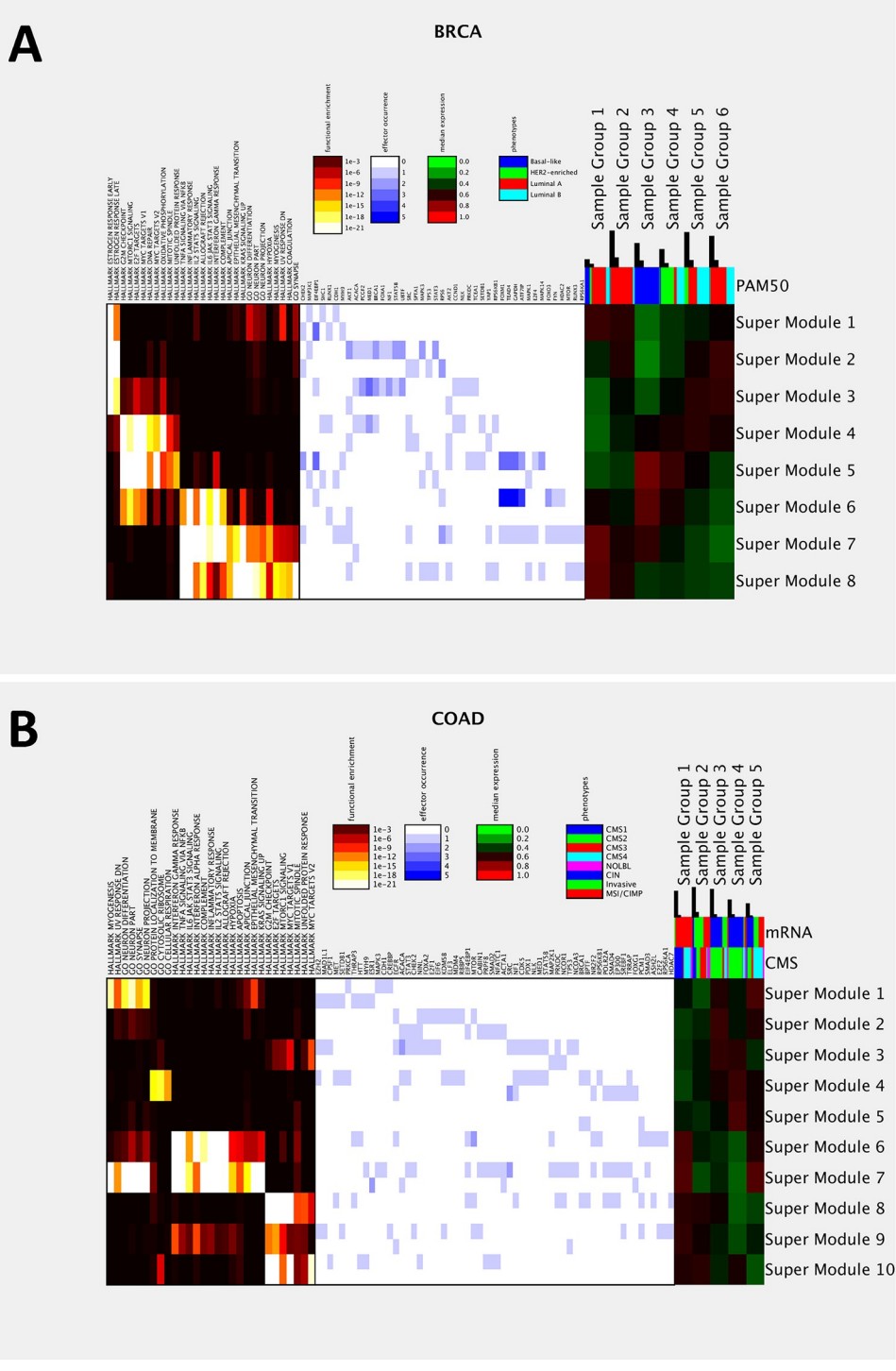

**Fig 5. IHAS integrated views in two cancer types. A**: An integrated view of BRCA data. The left panel displays the FDR-adjusted hyper-geometric enrichment p-values of selected Gene Sets in each Super Module. The middle panel displays the occurrences of selected effectors (hubs in the Consensus Artery Network) in each Super Module. The occurrences of positive and negative associations are shown in the upper and lower portions of each grid. The right panel displays the median combinatorial expression of each Super Module in each Sample Group. The colored bars above the right panel shows the composition of clinical phenotypes (PAM50 subtypes for BRCA) in each Sample Group. The black bars above indicate the total number of patients and the number of patients surpassing 5-year survival times in each Sample Group. **B**: An integrated view of COAD data. Legend follows panel **A**.

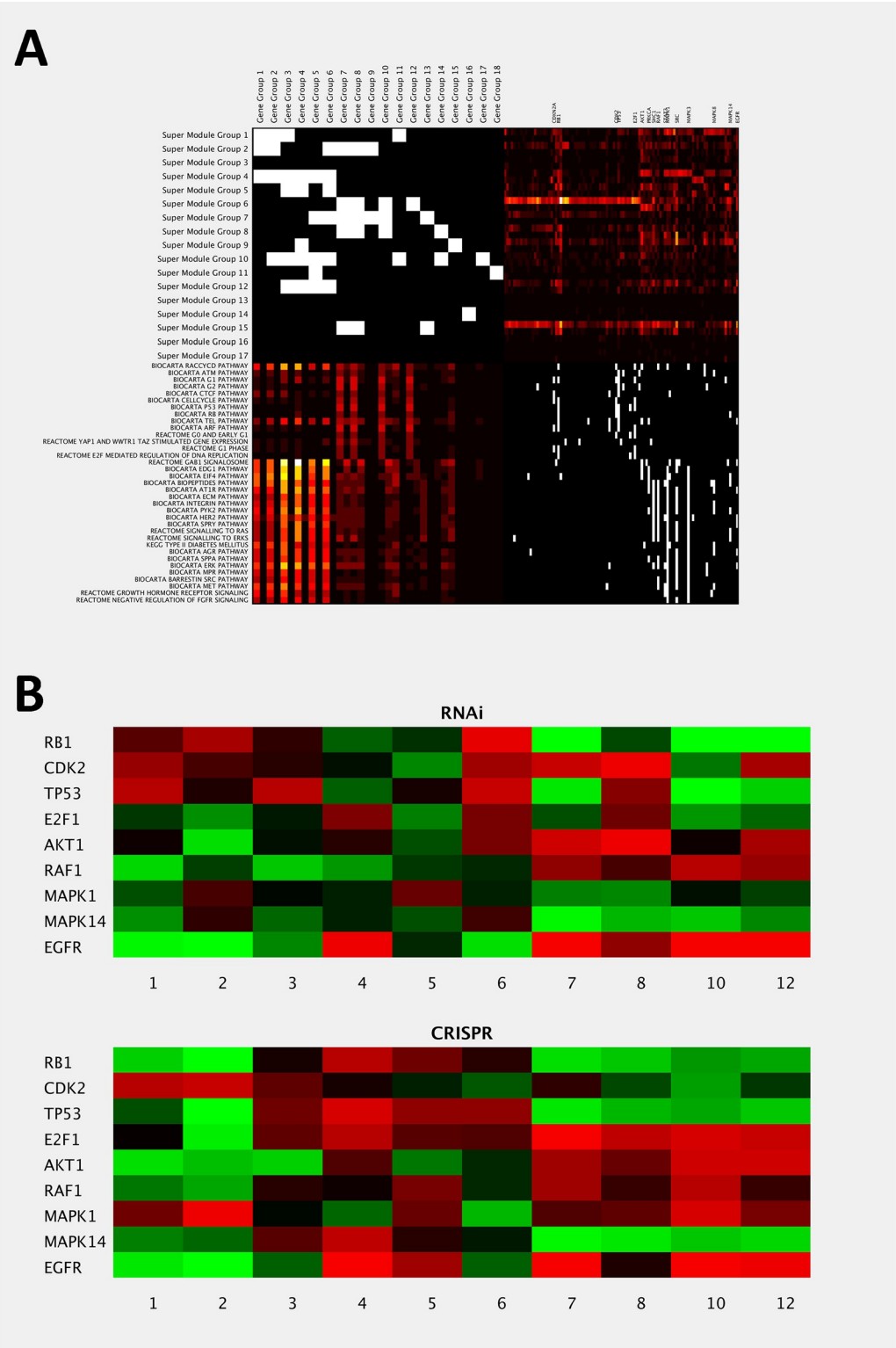

**Fig 6. IHAS integrated views in multiple cancer types. A**: An integrated view of recurrent hub effectors and the functions of their impacted target genes. It consists of four patches. The upper-left patch ($M_1$) shows the binary over-representation matrix of Gene Groups in Super Module Groups (S1E Table). The upper-right patch ($M_2$) shows the occurrence frequencies of 101 selected hub effectors in each Super Module Group. The counts of positive and negative associations are shown in the upper and lower portions of each grid. The lower-right patch ($M_3$) displays the membership

matrix of the selected hub effectors in 35 known pathways. The lower-left patch ($M_4$) displays the aggregate occurrence frequencies of effector pathways in Gene Groups (which resembles $M_1 \cdot M_2 \cdot M_3$). Positive and negative associations are separated as in $M_2$. The 35 pathways are subdivided into two groups in terms of their patterns in $M_4$. 16 hub effectors appear frequently in the pathways and are annotated on top of $M_2$. **B**: Validation of the impacts of perturbing 9 selected hub effectors in Achilles data. Each grid displays the normalized mean correlation coefficient between the Achilles dependency profile of perturbing each selected hub effector and the CCLE mRNA expression profiles of each Gene Group belonging to the three Meta Gene Groups. The upper and lower panels are derived from RNAi and CRISPR data respectively.

group 1 pathways and AKT1, PRKCA, SHC1, RAF, STAT3, MAPK1, SRC, MAPK3, MAPK8, MAPK14, and EGFR in group 2 pathways.

## Validation on external datasets

To justify generalizability of IHAS, we performed validations in numerous external datasets. In the first part, we manifested the veracity of IHAS by indicating that various aspects of IHAS–such as expression coherence of target genes, associations between effectors and targets and between IHAS subunits and clinical features–were preserved in external datasets of METABRIC, REMBRANDT, GEO and CCLE. In the second part, we exhibited the utility of IHAS by showing that the expression signatures of IHAS subunits were informative about drug treatment and gene perturbation responses of cancer cell lines in CCLE drug response data and Achilles gene dependency data. In the third part, we explored the origins of IHAS subunits by demonstrating that the IHAS signatures were also present in Bodymap transcriptomic and Roadmap epigenomic data of normal tissues.

**Combinatorial expression patterns of Super Modules in TCGA BRCA and GBM are preserved in METABRIC and REMBRANDT data.** Molecular Taxonomy of Breast Cancer International Consortium (METABRIC, [77]) and Repository of Molecular Brain Neoplasia Data (REMBRANDT, [78]) comprise mRNA expressions, CNV, subtype labels, and survival times of 1981 breast cancer primary tumors and 176 brain tumors respectively. The combinatorial expression patterns of sorted Super Modules and Sample Groups in TCGA BRCA and GBM and the aligned counterparts in METABRIC and REMBRANDT are highly similar (Fig 7 and S12C, S12D, S12G and S12H Fig, random permutation p-values are <0.001 and 0.009). Target gene expressions remain highly coherent ($p_{diff} \geq 0.3$, deviation between the mRNA expression correlation coefficient distributions of Super Module target genes and all genes in the data) for all Super Modules in METABRIC and all but one Super Modules in REMBRANDT (S7A and S8A Tables). The majority of the CNV-mRNA associations in TCGA are also moderately preserved ($p_{diff} \geq 0.1$) in METABRIC (54 of 61) and REMBRANDT (11 of 22) (S7B and S8B Tables). The orders of Kaplan-Meier curves of the Patient Groups are not persistent between TCGA and external data (S12A, S12B, S12E and S12F Fig). However, the directions and magnitudes of the Cox regression distributions in most Super Modules are not preserved in METABRIC but are preserved in REMBRANDT (S7C and S8C Tables).

**TCGA Super Modules are aligned with GEO cancer transcriptomic datasets.** We downloaded 294 cancer transcriptomic datasets covering nearly all 33 cancer types from the GEO database. For each GEO dataset, we (1) calculated expression coherence ($p_{diff}$) of each Super Module, (2) sorted and grouped samples and aligned them with Sample Groups in the corresponding TCGA data, (3) manually checked whether the order of the Kaplan-Meier curves of aligned Sample Groups was compatible with that in TCGA data, (4) checked whether the Sample Groups were aligned with their histology/subtype annotations.

S9 Table summarizes the basic attributes and validation outcomes of the 294 GEO datasets. 247 datasets retain coherent expressions ($p_{diff} \geq 0.1$) in more than half of their Super Modules.

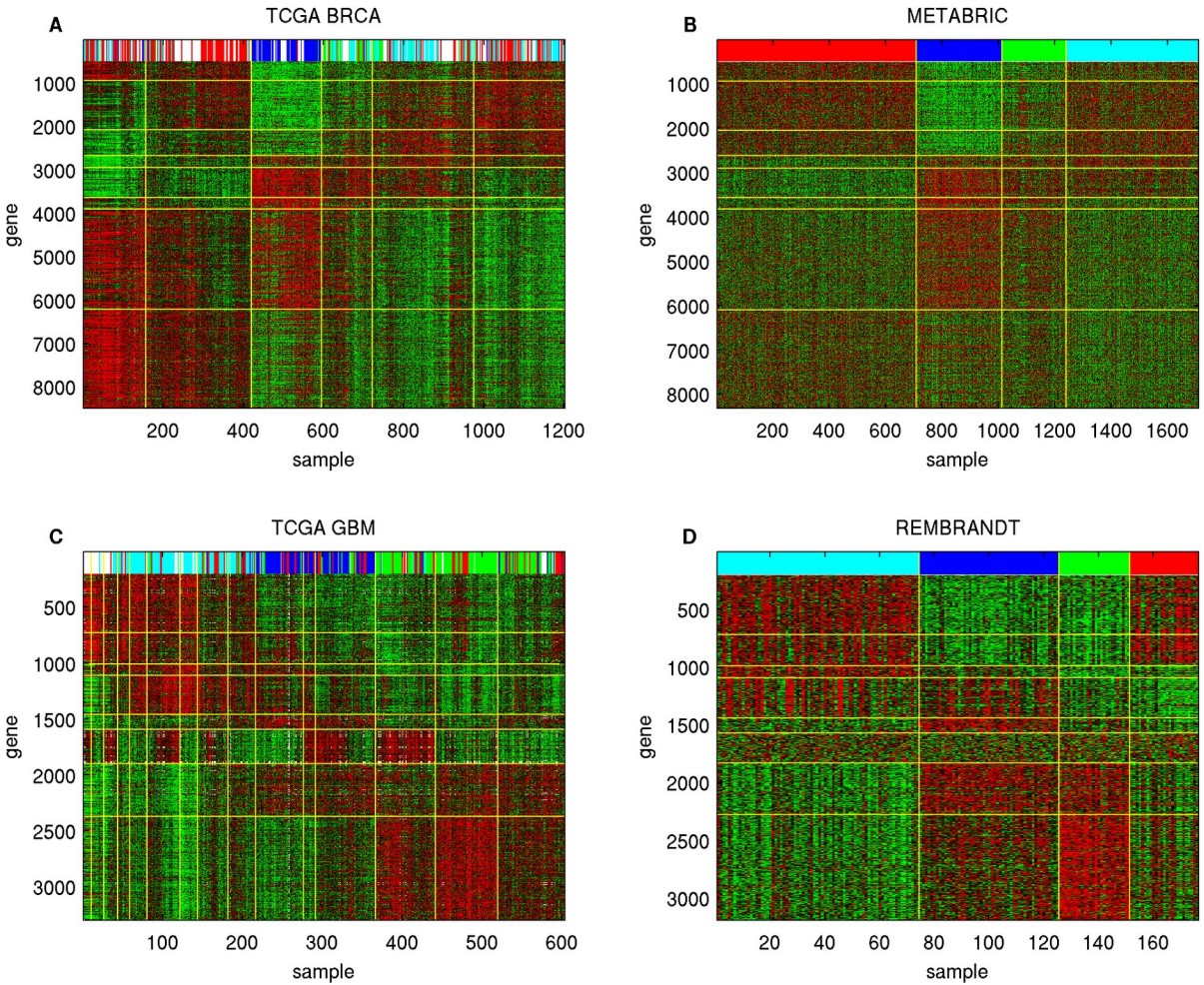

**Fig 7.** Sorted mRNA expression data of (**A**) TCGA BRCA and (**C**) GBM and the counterparts of (**B**) METABRIC and (**D**) REMBRANDT. Genes are sorted by their Super Module memberships and retain the same order in TCGA and external data. Samples are sorted by Sample Groups in TCGA data and by molecular subtypes in external data (luminal A, basal-like, HER2-enriched, and luminal B for METABRIC, and proneural, classical, mesenchymal, and neural for REMBRANDT). Sample subtypes are marked on the top row with the following codes: luminal A (red), basal-like (blue), HER2-enriched (green), and luminal B (cyan) for breast cancer, and proneural (cyan), classical (blue), mesenchymal (green), and neural (red) for GBM.

214 datasets possess the aligned expression patterns compatible with the TCGA data (incompatibility score ≤0.3). Among the 54 GEO datasets with survival times, the Kaplan-Meier curves of 18, 20, and 16 datasets are fully, partially and not compatible with TCGA datasets respectively. The aligned Sample Groups in 147 datasets roughly separate histological and/or molecular subtypes according to their annotations.

We also visualized the aligned combinatorial expressions and Kaplan-Meier curves (if survival information exists) of all GEO datasets accompanied with the corresponding TCGA data in Supplementary Data. Visualization is elucidated in a GEO dataset GSE68465 (Fig 8). GSE68465 is a lung adenocarcinoma dataset [79]. The combinatorial expression data and Kaplan-Meier curves of the aligned samples resemble those of the TCGA LUAD data (Fig 8A and 8C). Sample Groups are roughly compatible with tumor grades. Sample Groups 1, 2, 3, and 4 are enriched with well differentiated, moderately differentiated, moderately and poorly differentiated, and poorly differentiated tumors respectively, and their survival times follow a decreasing order (Fig 8B and 8D).

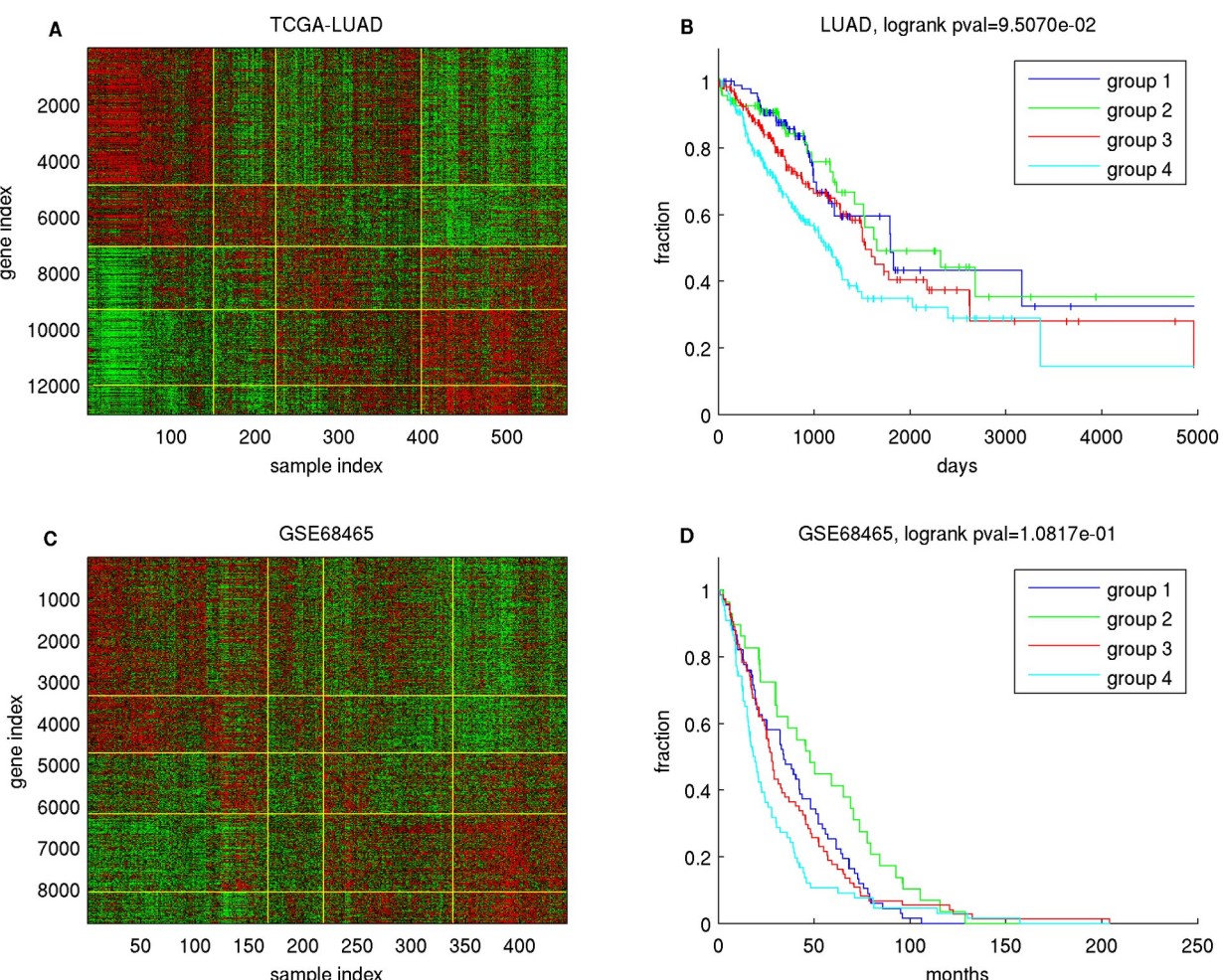

**Fig 8.** Sorted expression data and Kaplan-Meier curves of TCGA LUAD data (**A** and **B**) and the aligned counterparts (**C** and **D**) of a GEO lung cancer dataset (GSE69465). The four Sample Groups are separated by yellow vertical lines (**A** and **C**) and from left (group 1) to right (group 4).

**Effector-target associations are retained in CCLE multi-omics data, and Gene Group expression patterns are indicative about treatment/perturbation responses in CCLE and Achilles data.** Cancer Cell Line Encyclopedia (CCLE, [80]) consists of multi-omics data of 1046 cancer cell lines derived from 36 cancer types. Project Achilles provides a comprehensive cancer dependency map where the relative abundance of CCLE cancer cell lines by perturbing genes with RNAi [81] and CRISPR-Cas9 [82] are reported. Unlike tumor datasets, expression coherence and CNV-mRNA associations (S10A and S10B Table) are preserved primarily in Meta Gene Group 3 but not in Meta Gene Groups 1–2. About half of the CNV-mRNA associations for Meta Gene Group 3 are retained in CCLE, but less than one quarter of the associations for Meta Gene Groups 1 and 2 are retained.

Associations with other types of molecular alterations were verified by checking whether effectors possessing more associations in TCGA retained more compatible associations in CCLE (Fig 9A and S13 Fig and S10C–S10F Table). For mutations and DNA methylations, the numbers of compatible association occurrences (blue dots) quickly decline with ranks of association occurrences in TCGA, but the numbers of incompatible association occurrences (red dots) are far below those of compatible association occurrences at top ranks and decline very

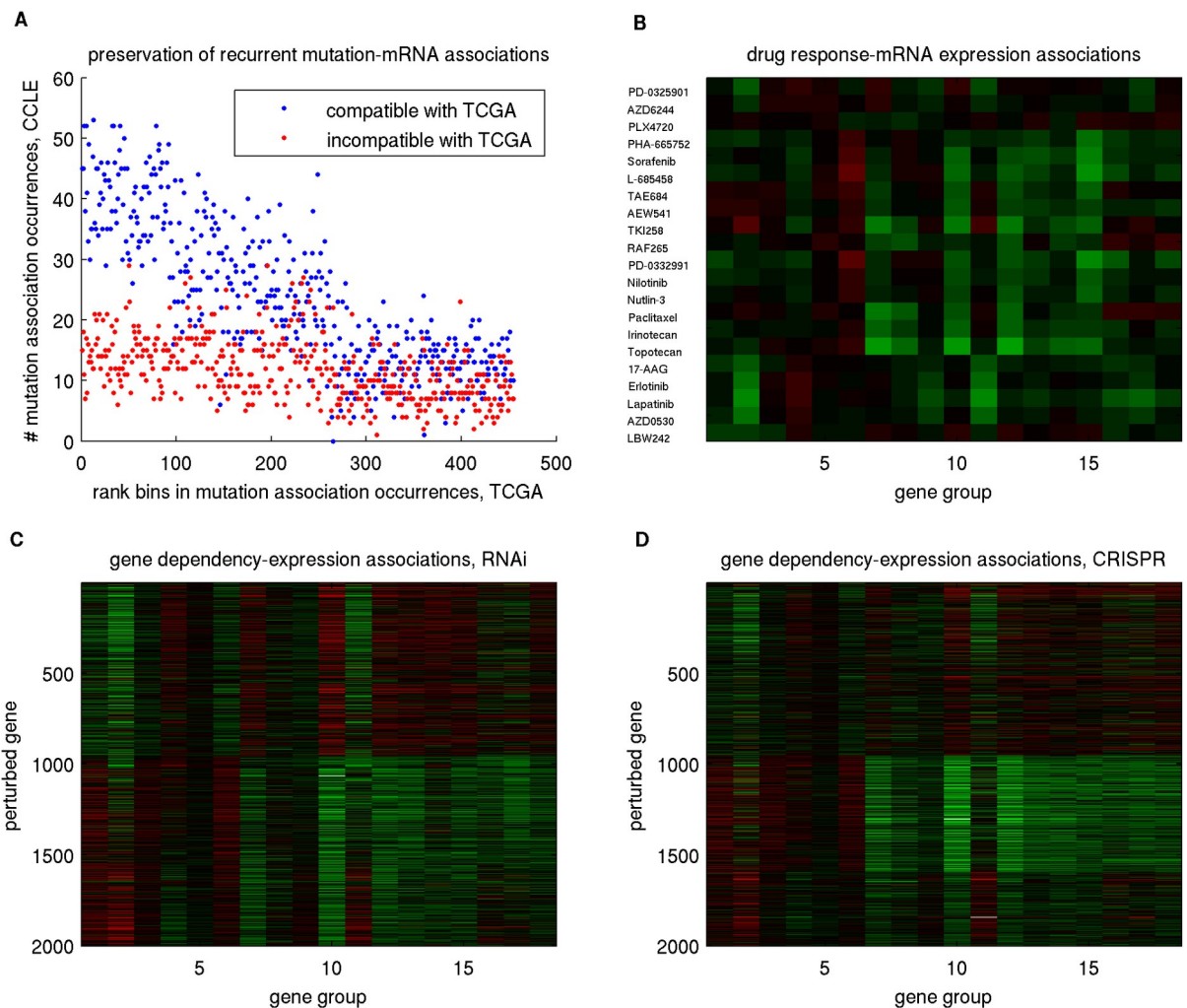

**Fig 9. Selected validation results on CCLE and Achilles data. A**: Sorted mutation-mRNA associations in TCGA by their occurrences and shows the occurrences of corresponding compatible (blue) and incompatible (red) associations in CCLE. **B**: Visualization of the drug response-Gene Group expression association strengths and directions in CCLE. Red and green entries indicate positive and negative associations respectively. **C** and **D**: The gene dependency-Gene Group expression association strengths and directions for RNAi and CRISPR perturbations in Achilles data.

slowly. In contrast, for microRNA expressions and protein phosphorylations the variations of the compatible and incompatible association occurrences are almost indistinguishable.

CCLE also comprises $IC_{50}$ values of 1046 cell lines' responses to drugs. An $IC_{50}$ value reports the dosage that reduces the cell line's population by 50%, hence a lower $IC_{50}$ value implies a more sensitive drug response to inhibit cell proliferation. We assessed associations between the response profiles of 21 drugs and the expression profiles of 18 Gene Groups over all cell lines (Fig 9B and S10G Table). Negative (green) and positive (red) associations denote that cell lines with high expressions of a Gene Group tend to be more and less sensitive to a drug treatment respectively. The 21 drugs are roughly subdivided into three groups. Drugs 1–3 are MEK or Braf inhibitors and have weak or no associations with all Gene Groups. Drugs 4–16 include all the cytotoxic agents and some kinase inhibitors and have strong negative associations with Meta Gene Group 3 and moderately positive associations with Gene Group 6. Drugs 17–21 are EGFR inhibitors and have moderately negative associations with Gene Groups 2, 9 and 11 and weak or no associations with other Gene Groups.

The Achilles project systematically identifies genes required for cellular vitality by performing high-throughput loss-of-function screening on CCLE cancer cell lines using RNAi and CRISPR-Cas9 technologies. A large negative value in a (gene,cell line) entry denotes the cell line population considerably declines upon the gene deletion. The dependency profiles of 15366 genes between two perturbation technologies are weakly correlated (S14A Fig, mean correlation coefficient 0.0463), but genes with strong positive correlation coefficients are highly enriched with the 459 cancer driver genes from the IntOGen database [83] (S14B Fig, KS p-value $2.45 \times 10^{-47}$). Thus we considered only the top 2000 genes in terms of the RNAi-CRISPR correlation coefficients.

We assessed the association (mean correlation coefficient) between the dependency data of each selected perturbed gene and the mRNA expression data of members in each Gene Group, sorted and subdivided the 2000 perturbed genes into 3 clusters (Fig 9C and 9D and S10H Table). The association patterns between the two perturbation technologies are highly similar. The gene dependency profiles of cluster 1 (genes 1–950) are negatively associated with mRNA expressions of Meta Gene Group 1 and positively associated with mRNA expressions of Meta Gene Group 3, and the gene dependency profiles of clusters 2 (genes 951–1594) and 3 (genes 1595–2000) have nearly opposite patterns from cluster 1. Gene Sets enrichment p-values of the three clusters are reported in S10I Table. Curiously, there is a mismatch between the enriched functions of perturbed gene clusters and Gene Groups. Cell lines with high expressions of Meta Gene Group 3 (enriched with cell cycle process) are more vulnerable to deletions of cluster 2 genes (enriched with respiration and translation). In contrast, cell lines with high expressions of Meta Gene Group 1 (enriched with immune response) or Gene Group 6 (enriched with cell junction) are more vulnerable to deletions of cluster 1 genes (enriched with 3 Meta Gene Groups).

In addition to validate IHAS associations at a group level, we also used the Achilles data to validate IHAS predictions pertaining to individual genes. Integrative analysis of pan-cancer effectors and targets indicates the hub effectors in two groups of pathways impact distinct Meta Gene Groups (Fig 6A). We identified 9 hub effectors which frequently occurred in selected Super Module Groups and pathways and retained moderate correlations ($\geq 0.1$) between RNAi and CRISPR data. The 4 hub effectors of pathway group 1 –RB1, CDK2, TP53, E2F1 –impact Meta Gene Group 3, thus the cell lines with higher Meta Gene Group 3 expressions are likely more dependent on these effectors. Similarly, the cell lines with higher Meta Gene Group 1 or 2 expressions are likely more dependent on the 5 hub effectors of pathway group 2 –AKT1, RAF1, MAPK1, MAPK14, and EGFR. We validated these predictions with the mean correlation coefficients between the Achilles dependency data of perturbing these effectors and the CCLE mRNA expression data in the three Meta Gene Groups (Fig 6B). To our satisfaction, group 1 effectors generally induce negative correlation coefficients (stronger dependency) in Meta Gene Group 3 members, and group 2 effectors generally induces negative correlation coefficients in Meta Gene Group 1 members. Specifically, RB1 and TP53 induce strong and consistent negative correlations in Gene Group 7, while AKT1 and EGFR induce strong and consistent negative correlations in Gene Group 2. Among the 15366 perturbed genes that appear in both RNAi and CRISPR data, 894 (5.82%) and 779 (5.07%) of them possess consistent negative correlations in Gene Groups 7 and 2 respectively.

**Super Modules and Gene Groups are enriched with tissue-specific genes in Illumina Bodymap data.** IHAS also contains information regarding the regulatory programs of normal tissues manifested in their transcriptomes and epigenomes. We verified IHAS on Illumina Bodymap, a transcriptomic dataset covering 27496 genes and 16 normal tissues (https://www.ebi.ac.uk/gxa/experiments/E-MTAB-513/Results). Meta Gene Groups 3, 1 and 2 possess strong, moderate and weak expression coherence respectively (S11A Table). We identified

8494 genes uniquely expressed in each normal tissue (S11B Table) and calculated the enrichment p-values of those tissue-specific genes in IHAS (S11C and S11D Table). Fig 10 displays the enrichment p-values of 16 Tissue-Specific Gene Sets in (A) 217 Super Modules sorted by 17 Super Module Groups with the same order as Fig 3A, (B) 217 Super Modules sorted by 33 cancer types, (C) 18 Gene Groups. Fig 10A and 10C indicate that several Tissue-Specific Gene Sets are enriched in IHAS compatible with enriched functions in Table 3 and S2B Table, such as leukocyte-specific genes in Super Module Groups 1–4 and Gene Groups 1–3 (enriched with immune responses), and testis-specific genes in Super Module Groups 6–8 and Gene Group 7 (enriched with cell cycle process). Fig 10B also reveals the tissues of origin of some Super Modules. Brain-specific genes are enriched in Super Modules of central nervous system cancers (PCPG, GBM and LGG). Similar enrichment links are observed in breast, colon, kidney, liver, lung, skeletal muscle, testis, and thyroid.

## Super Modules and Gene Groups are enriched with tissue-specific genes in Roadmap Epigenomic data

In addition to transcriptomes we also verified IHAS on a large epigenomic data of 129 normal tissues [84]. The raw data from several types of genome-wide assays are reduced to consecutive regions labeled with 25 epigenomic states (S12A Table). We further simplified epigenomic states into the binary states of potentially active or inactive transcription. Similar to the validation outcomes on CCLE and Bodymap data, Meta Gene Group 3 retains the strongest epigenomic coherence, while Meta Gene Groups 2 and 1 are partially coherent and incoherent respectively (S12B Table). We generated 8 gene clusters according to their epigenomic profiles (Fig 11A, S12C Table) and assessed their enrichment of each Super Module (Fig 11B and S12D Table) and Gene Group (Fig 11C and S12E Table). Each gene cluster possesses a distinct pattern of epigenomically active tissues: 1: all on, 2: all off, 3: neurons, 4: stem cells, 5: stem cells and neurons, 6: all tissue types but blood, 7: all tissue types except stem cells and blood, 8: blood. The enrichment patterns are compatible with the tissue-specific functions of TCGA Gene Groups. For instance, cluster 1 (all on) is enriched with Gene Groups 4, 10, 12, 15 (cell cycle and general transcription pathway), and cluster 3 (neurons) is enriched with Gene Groups 5, 18 (neuron development and projection).

## Comparison of IHAS with other multi-omics integration studies and databases

A number of recent studies integrated multi-omics data across multiple cancer types in TCGA. While these studies share similar characteristics of vertical and/or horizontal data integration with IHAS, each work possesses specific objectives, assumptions, approaches and focused biological processes. It therefore lacks a single yardstick to measure the performance of these methods in a unified framework. Instead, we compared IHAS with these methods by first qualitatively listing their features and second quantitatively evaluating the overlap or similarity of their analysis outcomes. We solicited 5 studies for comparison: iClusters of tumors [12], immune subtypes of tumors [54], Multi-Omics Factor Analysis (MOFA [55]), Multi-omics Master-Regulator Analysis (MOMA [56]), and tumor microenvironment subtypes (TME [57]). Besides individual multi-omics integrative studies, we also compared the IHAS inference results with the known and predicted protein-protein associations from the STRING database [85].

The presence or absence of 12 features in IHAS and 5 other methods are listed in S13A Table: (1) hierarchical representation of genes and tumors, subcategorization of (2) tumors and (3) genes, (4) molecular alteration signatures of tumor subcategories, (5) gene expression

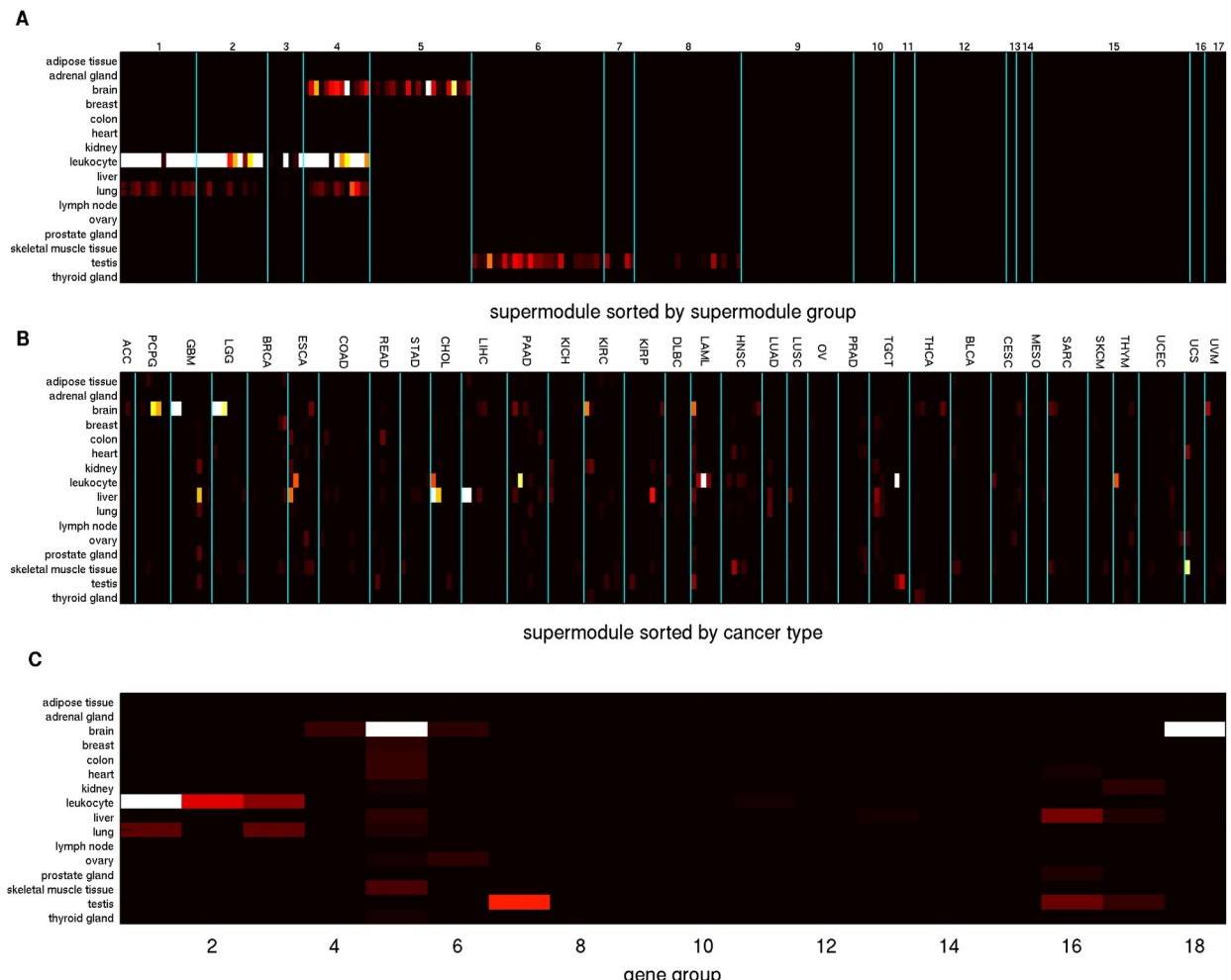

**Fig 10.** Enrichment p-values of tissue-specific genes derived from the Illumina Bodymap data in Super Modules sorted by **A**: Super Module Groups, **B**: cancer types, and in **C**: Gene Groups. Enrichment p-values of Super Modules and Gene Groups are reported in S11C and S11D Table.

signatures of tumor subcategories, (6) associations between molecular alterations and gene expressions, (7) multi-modal integration, (8) inclusion of known network data, (9) cancer type specific analysis, (10) pan-cancer analysis, (11) associations with clinical phenotypes, and (12) validations on external data. While IHAS possesses all 12 features, each reference method lacks at least one feature. iClusters do not explicitly subcategorize genes and build associations of genes. Immune subtypes manually pick and subcategorize genes and are based on mRNA signatures alone. MOFA neither handles horizontal integration across cancer types nor gives direct gene-level interpretations of the factorized matrices. TME subtypes are constructed by manually selecting 29 functional gene expression signatures alone. MOMA comprises the most comprehensive analysis including including 11 features. Nevertheless, none of the reference methods provides a hierarchical representation of the inferred subunits as for IHAS.

We also quantified the overlap or similarity of gene/tumor subcategories derived from IHAS and reference methods. 8 Pan-cancer Sample Groups from IHAS and 28 tumor iClusters exhibit significant overlap in only a few combinations (S13B Table). In contrast, 8 Pan-cancer Sample Groups are strongly aligned with 6 immune subtypes (S13C Table). MOMA tumor subtypes are not provided in their publication, and MOMA MRBs (gene subcategories) are

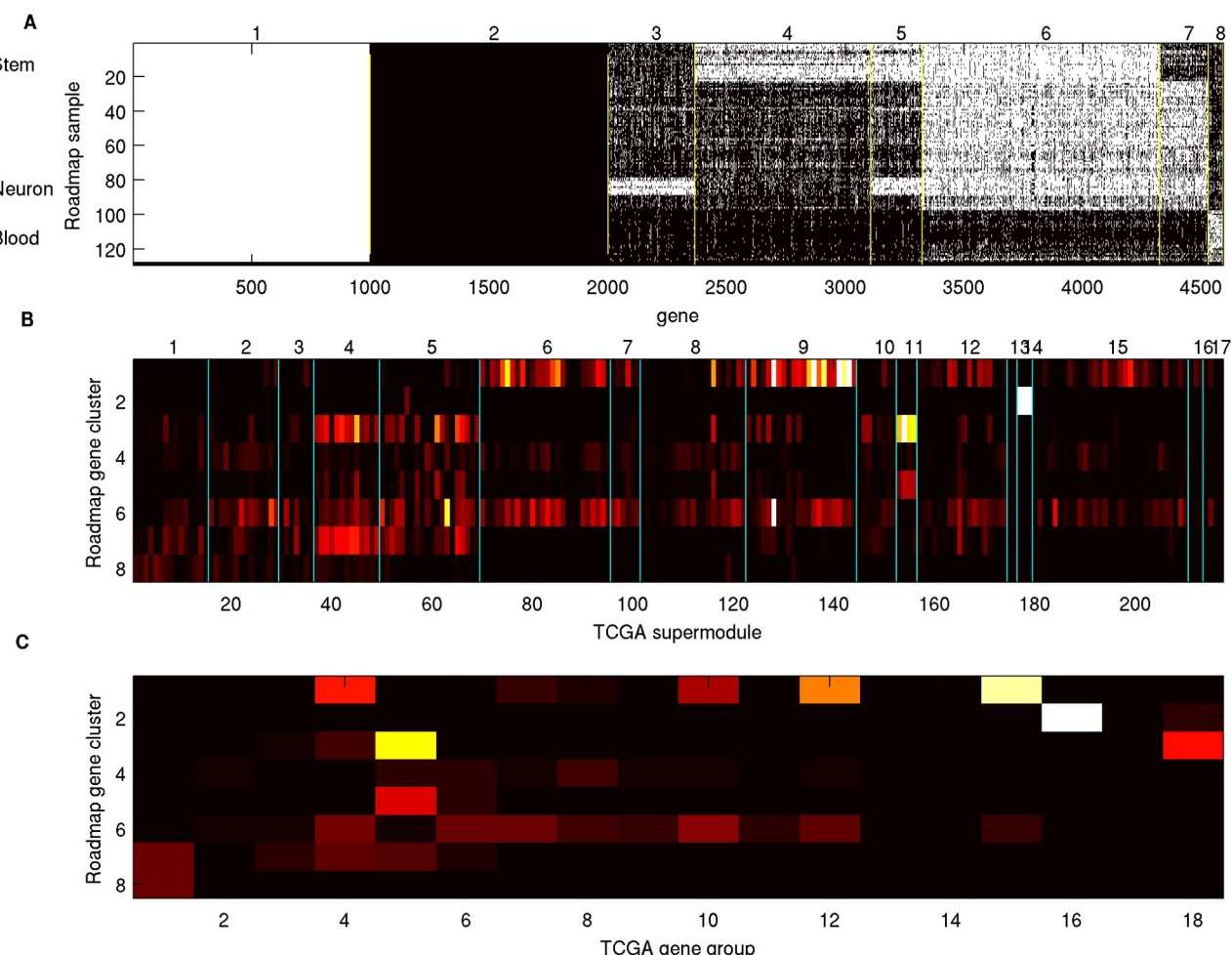

**Fig 11. A**: Combinatorial patterns of transcriptional regulation states of eight tissue-specific gene clusters from the Roadmap epigenomic data. **B**: Enrichment p-values of Roadmap gene clusters in TCGA Super Modules. **C**: Enrichment p-values of Roadmap gene clusters in TCGA Gene Groups. Enrichment p-values of Super Modules and Gene Groups are reported in S12D and S12E Table.

marginally overlapped with IHAS Gene Groups (S13D Table). TME gene signatures are significantly overlapped with only Gene Groups 1 and 7 (S13E Table). The 4 TME subtypes are strongly overlapped with some Pan-cancer Sample Groups (S13F Table). We also employed MOFA to the integrated data and compare the decomposition outcomes with IHAS Sample Groups in each cancer type. S13G Table reports the fractions and median enrichment p-values of overlap counts between aligned clusters for each cancer type. The overlap fractions are between 0.4 and 0.6 in most cancer types, and the median enrichment p-values are significant ($p \leq 10^{-5}$) in the majority of cancer types, suggesting that MOFA sample clusters are moderately related to IHAS Sample Groups.

Within some cancer types, the TCGA Consortium also reports sample clusters derived from three integrative algorithms–iclusters [86], clusters-of-clusters (COC [11]), and PARA-DIGM [39]. Sample Groups in IHAS are strongly or moderately aligned with those sample clusters: 6 of 7, 4 of 4, and 5 of 5 cancer types with icluster, COC and PARADIGM labels have concentration coefficients ≥0.5. The concordance between Sample Groups and those clustering outcomes is expected since they all capture the dominant combinatorial expression patterns and the associated effector molecular alterations within cancer types.

The mRNA expressions of IHAS exhibit complicated and variable relations with survival times. Across cancer types, directions and strengths of associations between Gene Group expressions and survival times (Fig 4D) resemble the prognostic concordance indices of the characteristic immune expression signatures ([54], Fig 3B). The lymphocyte infiltration and macrophage regulation signature resembles the signature of Meta Gene Group 1, while the wound healing signature resembles the signature of Meta Gene Group 3.

STRING is a large database of known and predicted protein-protein associations with confidence scores according to multiple lines of evidence. We validated the IHAS inference results on STRING by checking whether effector/regulator pairs and effector-target pairs co-occurring in the same Super Modules tend to possess high STRING scores. Gene pairs (effector-effector or effector-target pairs) were sorted by their co-occurring frequencies over the Super Modules. To assess enrichment in STRING we calculated the cumulative confidence scores (sum of the scores from the first to the current rank positions). As a negative control we randomly sampled the same number of effector/regulator pairs and calculated their cumulative scores. S15A and S15B Fig display the cumulative scores of effector-effector and effector-target pairs. The co-occurring pairs (blue curves) possess much higher cumulative scores than random pairs (red curves). We report the co-occurring frequencies and STRING scores of top 10000 effector/regulator and effector-target pairs in S13H and S13I Table respectively.

## Discussion

In this study, we built an Integrated Hierarchical Association Structure (IHAS) between molecular alterations on genomes/epigenomes and variations on transcriptomes in multiple cancer types. We inferred IHAS from the entire TCGA data and validated it in a wide range of external datasets. To our knowledge this is one of the most comprehensive characterization relating molecular alterations and transcriptomic variations in cancers. Below we discuss the clinical implications, limitations and potential extension of IHAS.

### Clinical implications

A major contribution of IHAS is that it organizes the rich and complex associations between genomic/epigenomic alterations and transcriptomic variations of cancers in a hierarchical structure. At a high level the transcriptomic variations of most tumors are reduced to the combinatorial patterns of three dominant biological processes (Meta Gene Groups): immune response, development and metastasis, and cell cycle control, as well as several other major processes (Gene Groups) such as translation and respiration. These combinatorial expression patterns are aligned with the majority of clinical and molecular features of cancers, such as PAM50 subtypes of breast cancer, CMS subtypes of colon cancer, mRNA and G-CIMP subtypes of GBM, IDH mutation status in low-grade gliomas, histology of sarcomas, and many others. When descending along the hierarchy, the common transcriptomic signatures are misregulated by diverse genomic and epigenomic alterations in different cancers, and other biological processes also come into play. This view concurs with the notion of cancer hallmarks [2] and several recent studies demarcating pan-cancer subtypes by omics signatures (e.g., [54,56,57,87]). IHAS provides not only this high-level picture but also the association information at multiple levels of details down to individual genes.

The combinatorial expression patterns of pan-cancer subunits in IHAS (Meta Gene Groups and Gene Groups) provide informative guidelines for targeted treatments. We have demonstrated that cancer cell lines with elevated Meta Gene Group 1 or 3 expressions are differentially sensitive to distinct sets of drugs (Fig 9B) and differentially dependent on perturbing two sets of effector genes (Fig 6B). Drugs or perturbations targeting cell cycle, DNA repair and

apoptosis are more effective in treating tumors which are proliferative but deficient of immune response and metastatic activities; drugs or perturbations targeting MAPK/ERK, MET, ECM, and various other signaling pathways are more effective in treating tumors which are metastatic and leukocyte infiltrated but less proliferative. Beyond this dichotomy, cell lines possessing elevated expressions in Gene Group 9 are sensitive to EGFR and BRAF inhibitors but independent of cytotoxic agents (Fig 9B). A possible connection between sensitivity to EGFR inhibitors and the enriched function of Gene Group 9 (enriched with protein synthesis) is through the PI3K-AKT pathway and mTOR [59,88]. Therefore, IHAS can assist precision cancer treatments by selecting the adequate drugs or identifying the candidate drug target genes based on combinatorial expression patterns of target genes and their associations with effector genes.

Although the combinatorial expression patterns of Meta Gene Groups and Gene Groups are ubiquitous across cancer types, their relations with patients' survival times are variegated. We have demonstrated that the directions of associations between survival times and Gene Group expressions vary with cancer types (Fig 4D). This property elicits caution when using transcriptional biomarkers to predict prognostic outcomes. For instance, high expressions of cell proliferation biomarkers (Meta Gene Group 3) indicate poor prognosis in breast cancer but good prognosis in colorectal cancer. Nevertheless, these relations still have recognizable patterns as Gene Groups of the same Meta Gene Groups possess similar $p_{diff}$ vectors, and the $p_{diff}$ vectors of the 33 cancer types are roughly subdivided into three groups.

Within a cancer type, IHAS supplies an integrative view of multiple aspects–combinatorial expressions of Super Modules and Sample Groups, functional enrichment of Super Modules, effectors possibly modulating the target gene expressions, and alignment of clinical features in Sample Groups (Fig 5 and S10 and S11 Figs). This integrative view facilitates understanding the underlying biological processes, stratifying patients, and identifying putative target genes for each patient stratum.

## Limitations of IHAS

Despite the rich information disclosed by IHAS, our analysis outcomes are limited in terms of data and algorithms. Recent studies (such as [89]) and this work implicate the importance of complex interactions of cancer cells and microenvironment in heterogeneous tumors. Single-cell sequencing data and the concomitant analysis are indispensable for investigating tumor-microenvironment interactions and mixtures of multiple cell types and subclones, but are not probed in TCGA. In addition, the TCGA data covers primarily exomes. The roles of alterations in non-protein coding regions are revealed in recent studies (e.g., [16]) but cannot be recapitulated in our work. These missing pieces can be possibly recovered by extending the IHAS framework to the lately published whole-genome cancer data by the PCAWG Consortium ([17–20]).

Besides data limitations the IHAS inference algorithm has several shortcomings. First, the model hypothesizes effector molecular alterations, target mRNA expressions, and downstream phenotypes are associated. This hypothesis does not always hold as some cancer drivers may affect phenotypes by altering gene activities/conformations/localizations without modulating transcription or translation [59], and mRNA quantities can be loosely coupled with protein quantities [90]. Second, a significant effector-target association requires a substantial occurrence frequency of the molecular alterations in an effector in order to reach sufficient statistical power. Genes undergoing rare alterations are thus not incorporated in the Association Models. Third, we generated 8 Pan-cancer Sample Groups in terms of the binary combinatorial expression patterns of three Meta Gene Groups. This crude-level characterization ignores the subtle

structures pertaining to other Gene Groups involved in respiration, protein synthesis, and other functions.

## Prospect

The hierarchical representation of associations is crucial for multi-modal, multi-cohort data integration in the biomedical context. Therefore, the IHAS framework can be in principle extended to analyze other large-scale integrated disease data. In practice, few other diseases exhibit diversity and attract intensive public and research attention comparable to cancer. Applying the IHAS framework to another multi-omics cancer data generated by the ICGC and PCAWG Consortium [17–20] is an obvious extension as they share very similar nature with TCGA data. Beyond cancer, another potential target for IHAS is a collection of neurological disorders since (1) many neurological disorders exhibit genomic, epigenomic and transcriptomic abnormalities, (2) multi-omics data of several neurological disorders (such as Alzheimer's disease and Parkinson's disease) are already available, and (3) different neurological disorders may share common molecular aetiology mechanisms and possess unique mechanisms similar to the relations between different types of cancers. Nevertheless, to our knowledge no consortium is formed to systematically collect multi-omics data of multiple neurological disorders. Extensive efforts to compile and standardize the data from diverse sources are required before vertical and horizontal data integration can be performed.

## Methods

### Ethics statement

The project underwent the following Institutional Review Board (IRB) review process in order to access the METABRIC data of breast cancer CNV, transcriptomes, and clinical information. Nevertheless, all the data used in this work are from public sources. We did not directly collect any data from patients.

Name of the IRB Committee: IRB on Biomedical Science Research, Academia Sinica.
Approval number: AS-IRB03-110399.

### Summary

The data processing and analysis algorithms in this study are divided into six sections: (1) overview of hierarchical relations of IHAS subunits, (2) collection and processing of the data, (3) inference of IHAS from TCGA data, (4) functional characterization of IHAS, (5) alignments of IHAS with clinical phenotypes, (6) validation of IHAS on external data. We give a brief sketch of some key algorithms below and report their detailed procedures in Supplementary Text S1.

### Hierarchical relations of IHAS subunits

Table 2 summarizes the IHAS subunits, and Fig 1G gives an overview of their relations. There are three chains of inclusion relations (vertical unidirectional lines in Fig 1G). In the first chain, Association Models between effector alterations and target gene expressions are grouped into Modules by common effectors; Modules within the same cancer type are grouped into Super Modules by similar combinatorial expression patterns; and Super Modules across cancer types are grouped into Super Module Groups by their shared member genes. In the second chain, genes are clustered to Gene Groups by the gene membership occurrence matrix of Super Modules (Fig 3A), and Gene Groups are clustered to Meta Gene Groups by common functional enrichments. In the third chain, samples in the same cancer type are

clustered to Sample Groups by the combinatorial expression patterns of Super Modules, and Sample Groups across cancer types are clustered to Pan-cancer Sample Groups by the combinatorial expression patterns of the Meta Gene Groups. Some subunits at the same level also form combinatorial relations. Super Modules and Sample Groups of the same cancer type form combinatorial expression patterns of certain biological processes in sample subtypes. Super Module Groups and Gene Groups together partition the gene membership occurrence matrix of Super Modules. Meta Gene Groups and Pan-cancer Sample Groups form combinatorial expression patterns of three major biological processes across all cancer types. The procedures of constructing these subunits and specifying their relations are sketched below and described in Supplementary Text S1. For instance, in the subsection "Inferring IHAS from TCGA data–Association Models", we depict a stepwise regression procedure to construct logistic regression-like Association Models; in the subsection "Inferring IHAS from TCGA data–Super Modules and Sample Groups", we depict a procedure combining spectral clustering and boundary detection to simultaneously cluster Association Modules and samples into Super Modules and Sample Groups respectively.

## Collecting and processing data

**Collecting TCGA and external data.** We downloaded the level-2 and level-3 data of all 33 cancer types from the TCGA data portal and the data of inferred purity [74] and stemness [75] of TCGA samples. We also compiled a unified network database of human biomolecular interactions from the following sources of biological pathways and networks: (1) Pathway-Commons ([91], version 4), (2) TRANSFAC human transcription factors and their target genes ([92], proprietary version 2009.1), (3) MiRTarBase of microRNA-target pairs ([93], version 4.5), (4) ENCODE data ChIP-Seq experiments in human cell lines ([94], version 2). The unified network is a hypergraph consisting of 90122 molecules (nodes) and 1068050 interactions (hyper-edges).

To characterize the functions of IHAS we downloaded 14545 Gene Sets from the MSigDB database [65]. We collected external omics datasets from the following sources to validate IHAS: (1) METABRIC data of 1981 breast cancer patients [77], (2) REMBRANDT data of 176 brain tumor patients [78], (3) 294 cancer transcriptome datasets, survival times and histological labels (for a subset of them) deposited in the GEO database, (4) CCLE multi-omics data of 1046 cancer cell lines covering all TCGA data types and responses to 21 (reduced from 24) drugs [80], (5) Achilles data of gene dependency on CCLE cancer cell lines [81,82], (6) Illumina Bodymap of transcriptomic data from 16 normal tissues (EBI Expression Atlas, E-MTAB-513), (7) Roadmap epigenomic data from 129 normal tissues [84].

**Processing the data.** We adopted the following normalization procedures to convert all types of omics data into the same format with compatible scales. The beta values of the DNA methylation in each gene promoter and the cumulative distribution function (CDF) values of mRNA, microRNA expressions, protein expressions and phosphorylations of each gene were converted into trinary probability vectors $(p(x = -1), p(x = 0), p(x = +1))$ of hidden states by *probabilistic quantization* [40]. The discrete values of mutation and SNP data (for mutation data, 0, 1, and 2 denote silent, missense and nonsense mutations of genes; for SNP data, 0, 1 and 2 denote homozygote major alleles, heterozygote alleles, and homozygote minor alleles) were converted into probability vectors with the entire mass concentrated in reported discrete states. Probabilistic quantization of probe-level CNV data was adjusted to fit the empirical distribution of amplification and deletion events, and probe-level CNV probability vectors were merged to segment-level CNV data.

## Inferring IHAS from TCGA data

**Association models.**   The associations of a target gene are specified by an exponential family model in Eq 1. S2B Fig illustrates the procedures of inferring the Association Models. In brief, it consists of four steps. First, the foundation of Association Model construction is hypothesis testing of two nested regression models, including parameter estimation, evaluation of log-likelihood ratios, $\chi^2$ and permutation p-values. Second, we performed associations between all pairs of molecular alterations and gene expressions and identified the candidate effectors for all target genes. The threshold values of test statistics scores were determined by numbers and rates of false discoveries. Third, we employed a stepwise regression-like algorithm to incrementally select the effectors that provide the highest additional explanatory power given the existing model. Fourth, before executing model selection we prioritized the candidate effectors of each target gene with first their shortest path distances to the target gene in the unified molecular interaction network and second their types of molecular alterations. All inference computations, including pairwise and full associations, were executed in a Dell Precision 7920 Tower Workstation, which allows parallel computations in 96 nodes.

**Association modules.**   Association Modules were immediately determined from the Association Models of all target genes. For each effector, we identified the target genes ($y$'s) of all the Association Models whose effectors ($x$'s) contained it. For trans-acting CNV and SNP associations, we also identified the regulators which constituted cis-acting associations between the effector and the regulator mRNA expression and trans-acting associations between the regulator mRNA expression and target gene mRNA expressions.

**Super Modules and Sample Groups.**   We jointly clustered Modules and samples according to the module target expression data. S16 Fig illustrates the joint clustering algorithm comprising three phases. First, we applied spectral clustering [95] recursively to binary partition and sort Modules and Samples separately. Second, we employed an algorithm in computer vision to detect boundaries [96] in the two-dimensional sorted expression data of Modules and Samples. The expression data of sorted Modules and Samples was smoothened by convolving with Gaussian kernels of varying standard deviations. The boundaries of a smoothened data were acquired by the zero-crossing points in its Laplacian. Finally, the boundaries derived from prior phases were reconciled. The boundaries which were robust against the choice of smoothing scales were selected.

**Super Module Groups and Gene Groups.**   The membership relations of genes in Super Modules constitute a discrete-value matrix $M$ of genes (rows) and Super Modules (columns), where an entry $M_{ij}$ denotes the number of modules which contain gene $i$ and belong to Super Module $j$ (Fig 3A). We proposed an algorithm to sequentially cluster Super Modules and genes from $M$ to form Super Module Groups and Gene Groups. First, we applied hierarchical clustering to Super Modules according to their Jaccard similarities. Boundaries of Super Module Groups were determined by the enrichment patterns of six functional categories or overlap of member genes. Conceptually, we identified the highest-level nodes in the dendrogram where the descendant Super Modules in both branches have similar enrichment patterns in the following functional categories: cell cycle, immune response, cell adhesion, ribosome, respiration, and synapse. The descendant Super Modules of these nodes constituted Super Module Groups. We used these functional categories in the stopping criteria for hierarchical clustering because we observed frequent enrichment of these functional categories in the majority of Super Modules. Second, genes were clustered according to their membership patterns of Super Module Groups. A gene was assigned to a Super Module Group if the number of constituting Modules containing the gene was significantly higher than the number derived from a null model of random assignments. Each gene accordingly possessed a binary vector of Super Module

Group memberships. We then retrieved the unique combinatorial patterns of Super Module Group memberships and sorted them by the numbers of their constituting genes. The genes of the top-ranking Super Module Group membership patterns form the Gene Groups.

## Characterizing functions of IHAS

**Building the Artery Networks spanned by explanatory paths for associations.** We adopted a network diffusion model to evaluate edge weights in the unified molecular interaction network as illustrated in S2C Fig. The connecting paths of all effector-target association pairs cover a large portion of the unified network and likely contain many spurious paths and links. To extract the most informative part from the massive number of connecting paths, we introduced the following simplifying procedures. First, for each association pair the connecting paths are weighted according to their lengths and topology. Short paths have higher weights than long paths according to parsimony. Paths traversing highly connected hubs have lower weights since they are more likely to form by chance. Second, we incorporated these two path weighting criteria and developed a network diffusion model to assign weights of unified network edges. Third, we developed an algorithm to identify the Artery Network spanned by high-weight edges. Fourth, we extracted the common portion of the Artery Networks across multiple cancer types and reported the Consensus Artery Network.

## Relating IHAS with clinical phenotypes

**Prognostic analysis of Sample Groups within cancer types.** The survival or censoring time (in days) of a patient is the interval from the date of first diagnosis to the date of reported death or last diagnosis respectively. We quantified associations between subunits in IHAS (Super Modules, Gene Groups) and survival/censoring times with three approaches. First, we evaluated the Cox regression coefficient of the mRNA expression profile of each gene in the data [97] and compared their distributions between the member genes in a subunit and all genes in the data. The p-values of non-parametric tests such as Kolmogorov-Smirnov and Mann-Whitney tests are highly sensitive to the number of genes and often approximate 0 even for a very small deviation between the two distributions. We proposed a novel measure $p_{diff}$ of the deviation between the two distributions, and reported the mean Cox regression coefficient and $p_{diff}$ score for each Super Module and Gene Group. Conceptually, we constructed random variables $X_1$ and $X_2$ whose PDFs were $p_1$ and $p_2$ respectively. $p_{diff}$ was the difference between two probabilities that $X_1$ is greater and smaller than

$X_2$ : $p_{diff} \equiv \Pr(X_1 > (X_2 + \epsilon)) - \Pr(X_1 < (X_2 - \epsilon))$, where $\epsilon$ is a small value. Second, for each subunit we subdivided patients into two groups according to their median expression levels, and calculated the log rank p-value of the Kaplan-Meier curves of the two groups [98]. Third, for each cancer type, we derived Patient Groups from Sample Groups, visualized their Kaplan-Meier curves, and manually constructed a decision tree that related their combinatorial expression patterns to survival times.

## Validating IHAS on external datasets

**Aligning the combinatorial expression patterns in the GEO datasets.** We proposed an algorithm to align the combinatorial expression patterns of each GEO data to the TCGA transcriptome data of the corresponding cancer type. Samples were aligned between the two datasets. We applied spectral clustering recursively to both datasets and generated two binary partition trees of samples. The two partition trees were aligned by dynamic programming to minimize the mismatch of aligned expression patterns and respect the topologies of both trees.

Samples in the GEO data were sorted according to the alignment outcome. Some GEO datasets also possess survival data and/or sample annotations such as tumor histology and stages. We visually compared the annotations in the aligned Sample Groups and inspected the Kaplan-Meier curves of the aligned Patient Groups in the two datasets and determined whether their orders completely, partially or did not agree.

### Relating Gene Group expressions and gene dependencies

The Achilles project reports the growth responses of selected CCLE cell lines by perturbing 15366 genes with RNAi and CRISPR technologies. We sorted the perturbed genes by their correlation coefficients between their RNAi and CRISPR dependency data and selected the top 2000 genes. To verify whether the expressions of Gene Groups are indicative about gene dependency responses, we (1) calculated the correlation coefficient between each pair of gene dependency and mRNA expression data, (2) obtained the average correlation coefficient over the mRNA expressions of genes in each Gene Group, (3) clustered the 2000 perturbed genes according to their average correlation coefficients from RNAi and CRISPR data separately. The sorted average correlation coefficients are visualized in Fig 9C and 9D, sorted genes are reported in S10H Table, and enriched functions are reported in each cluster in S10I Table.

### Relating IHAS from TCGA and transcriptomic and epigenomic data in normal tissues

We also verified IHAS in the transcriptomic and epigenomic datasets in normal tissues (Illumina Bodymap and Roadmap data respectively). For Bodymap data, we identified the genes which were uniquely expressed in one of the 16 tissues (S11B Table). For Roadmap data, we extracted the segment data labeled with 25 predicted epigenomic states (S12A Table), and reduced the data into the binary states of active transcription of each gene. Tissues were roughly subdivided into four groups–stem, neuron, blood, and others. Genes were categorized into 8 clusters according to their overall binary states of active transcription over the four tissue groups.

To verify IHAS in Bodymap and Roadmap data, we examined whether genes possessing tissue-specific expression or epigenomic states were strongly enriched in Super Modules and Gene Groups. Standard hyper-geometric enrichment p-values of tissue-specific genes in Bodymap and Roadmap in Super Modules and Gene Groups were calculated and reported (Figs 10 and 11).

We place the IHAS inference and validation results in Supplementary Data organized as hierarchical Webpages and deposited in the Synapse database as a zip file with the link synapse.org/#!Synapse:syn30165761/files/IHAS_data.zip. We also place the Webpages under https://www.stat.sinica.edu.tw/IHAS/. It consists of the data of the following five categories.

1. Data processing. Intermediate results of data processing, including CNV segments of each cancer type and chromosome, and the false discovery characteristics for determining threshold values for pairwise associations.

2. Integrated Hierarchical Association Structure (IHAS). Content of IHAS, including models, modules, Super Modules, Super Module Groups and Gene Groups.

3. Functional characterization. Three functional characteristics of IHAS, including Gene Set enrichment outcomes, recurrent effectors, and the artery networks explicating the effector-target associations.

4. Associations of IHAS with clinical phenotypes. Two aspects of IHAS-phenotype associations, including Sample Groups and alignments with molecular subtypes and features, and associations with survival times.

5. Validation on external datasets. Validations of IHAS on seven external datasets, including METABRIC for breast cancer, REMBRANDT for brain tumors, transcriptomic datasets from GEO, CCLE cancer cell lines and Achilles dependency map, connectivity map of drug responses, Illumina Bodymap for normal tissue transcriptomes and Roadmap epigenomic data of normal tissues.

## Source codes

Source codes of 20 programs written in C and Matlab and example data extracted from open-layer TCGA datasets are deposited in the Synapse database as a zip file with the link synapse.org/#!Synapse:syn30165761/files/IHAS_programs.zip. These programs cover major procedures of generating the inference and validation results in this study. They include the programs of data processing (such as partitioning chromosomes into CNV segments), constructing IHAS (such as building Association Models and combining them to Association Modules and Super Modules), functional characterization (such as evaluating edge weights by a network diffusion model), associations with clinical phenotypes (such as survival analysis), and validations on external data (such as aligning IHAS target gene expressions between TCGA and GEO data). Example data from TCGA and external sources are also included.

## Supporting information

**S1 Fig. Architecture and information flows of the IHAS inference machine.** The single cancer type data analysis (the top box) is undertaken for each cancer type separately. In each cancer type the TCGA omics data are processed and fed into the model inference algorithm to build Association Models and Association Modules. The Association Modules and mRNA expression data are bi-clustered to form Super Modules and Sample Groups. The (effector,target) pairs in the Association Models are used to construct the Artery Network from a unified biomolecular network. The Super Modules and Sample Groups then undergo subtype alignments, functional enrichment, and prognostic associations. The inference outcomes of individual cancer types are integrated to form pan-cancer subunits (Super Module Groups, Gene Groups, Consensus Artery Network). These pan-cancer structures are aligned with pan-cancer phenotypes (molecular phenotypes and prognosis), characterized (Recurrent Effectors, functional enrichment, and hubs in the Consensus Artery Network), and validated on external data (individual cancer types, perturbations, normal tissues). Black lines indicate TCGA data and association outcomes. Red lines indicate external data for validation. Blue lines indicate non-TCGA data used to infer IHAS.
(TIFF)

**S2 Fig. Schematic diagrams of hierarchical association structure inference and characterization. A**: Directions of associations according to the central dogma view. Alterations on genomes (sequence mutations, copy number variations, single nucleotide polymorphisms), epigenomes (DNA methylations), microRNA expressions and protein phosphorylations modulate mRNA transcriptions. Variations of mRNA transcriptions modulate protein expressions. Transcriptomic and proteomic variations affect phenotypic variations. **B**: Procedures of building Association Models of individual genes. First, we calculate pairwise associations between all candidate effectors and targets and select the pairs whose strength of associations

(quantified by log-likelihood ratios and permutation p-values) pass threshold values (red edges in the top panel). Second, for each target gene candidate effectors are incrementally selected according to statistical hypothesis test outcomes. In a toy example in the middle panel, suppose we consider merging a candidate effector $M_1'$ with the current model $M_1$ to become the augmented model $M_2$, and $M_0$ is an independent model without effectors. We incur three hypotheses tests comparing pairs of nested models and select models by the three p-values according to the table on the right. Third, before incurring statistical model selection candidate effectors are prioritized by their shortest path lengths to the target gene in the molecular interaction network. Candidate effectors with shorter path lengths have higher priorities. Candidate effectors with the same path lengths are prioritized by their types: CNV > mutation > DNA methylation > microRNA expression > protein phosphorylation > SNP. A toy example in the bottom panel illustrates these priorities. **C**: Illustration of evaluating edge weights by a network diffusion model. Suppose there is an association between effector $s$ and target $t$, which are connected by paths in the molecular interaction network. The network diffusion model starts at $s$, iteratively jumps to downstream neighbor nodes with equal probabilities, and stops when reaching $t$ or other bottom nodes. The un-normalized weight $q(\pi)$ of a path $\pi$ is the probability of traversing along $\pi$ in random walks. The normalized weight $p(\pi)$ is the conditional probability of traversing along $\pi$ given that the random walker reaches $t$. The weight of an edge $e$ contributed from an $(s, t)$ pair is the sum of normalized path weights for all paths traversing $e$, which is the conditional probability of traversing $e$ given that the random walker reaches $t$. Finally, the total weight of an edge is the sum of edge weights contributed from all (effector,target) association pairs.
(TIFF)

**S3 Fig. The log$_{10}$ histograms of effector numbers for Association Models in each cancer type and all cancer types combined.** To avoid displaying $-\infty$ ($\log_{10} 0$) values we replaced zero counts with 0.001.
(TIF)

**S4 Fig. Variations of selected effectors and target gene expressions of four Super Modules.** **A**: BRCA Super Module 5. **B**: COAD Super Module 7. **C**: LGG Super Module 7. **D**: LIHC Super Module 2. Selected effectors are annotated on the upper half rows in each panel. The target genes are on the bottom half rows and not annotated. Color codes follow Fig 3A–3C.
(TIFF)

**S5 Fig. The whole Consensus Artery Network comprising indispensable interactions for explaining (effector,target) association pairs across multiple cancer types.** The genes with high connectivity are annotated. Node colors denote hub levels: 1 (red), 2 (purple), 3 (green), 4 (magenta).
(TIFF)

**S6 Fig. The information gain when moving downward along the hierarchy of IHAS. A**: Excerpts of four Super Modules enriched with the Gene Sets not belonging to the 18 Gene Groups and their functional enrichment in 13 selected Gene Sets representing the three Meta Gene Groups and several other functional processes. **B**: Correlation coefficients distribution of the mean target gene expressions between Association Models and the Super Modules they belong to. **C**: Correlation coefficients distributions of target gene expressions between the Association Models with identical effectors (black) and the Association Models sharing one common effector but also possessing distinct effectors (red). The $y$ axis in panels **B** and **C** indicates values from kernel density estimation.
(TIF)

**S7 Fig. The combinatorial expressions of Super Modules and Sample Groups overlaid with selected sample feature values in four cancer types. A**: BRCA, **B**: COAD, **C**: LGG, **D**: SARC. Dominant feature values of Sample Groups are reported in S5C Table. Feature values of individual samples are reported in S5D Table.
(TIF)

**S8 Fig. The prognostic analysis outcomes of three cancer types: BLCA, KIRC and LGG.** Left panels (**A**, **D**, **G**) display the combinatorial expressions of Super Modules and patient groups. Middle panels (**B**, **E**, **H**) display the Kaplan-Meier curves of patient groups. Right panels (**C**, **F**, **I**) display the decision trees segregating patient groups according to combinatorial expression patterns of Super Modules. Patient groups are separated by cyan vertical lines in their expression data (**A**, **D**, **G**) from left to right, and are annotated by the same colors in their survival curves (**B**, **E**, **H**) and decision trees (**C**, **F**, **I**).
(TIF)

**S9 Fig. The prognostic analysis outcomes of three more cancer types: LIHC, UCEC and UVM.** Legend follows S8 Fig.
(TIF)

**S10 Fig. IHAS integrated views in four cancer types: ACC, BRCA, COAD, ESCA.** Legend follows Fig 5.
(TIFF)

**S11 Fig. IHAS integrated views in four cancer types: LGG, LUSC, PCPG, SARC.** Legend follows Fig 5.
(TIFF)

**S12 Fig.** The Kaplan-Meier curves of Sample Groups and combinatorial expressions of Super Modules and Sample Groups of TCGA BRCA (**A** and **C**), METABRIC (**B** and **D**), TCGA GBM (**E** and **G**), and REMBRANDT (**F** and **H**) data.
(TIFF)

**S13 Fig. Preservation of recurrent TCGA associations in CCLE data.** Four panels report the analysis outcomes of four types of effectors: **A**: mutations, **B**: DNA methylations, **C**: microRNA expressions, **D**: protein phosphorylations. Legend follows Fig 9A.
(TIF)

**S14 Fig. Consistency between the gene dependencies with RNAi and CRISPR perturbations in Achilles data. A**: The distributions of correlation coefficients between gene dependency data by RNAi and CRISPR perturbations on the same genes. **B**: The enrichment outcomes of cancer driver genes in the top-ranking perturbed genes in terms of the correlation coefficients between RNAi and CRISPR perturbation data. The GSEA gap and its Kolmogorov-Smirnov p-value are reported.
(TIF)

**S15 Fig. Validation of IHAS inference results on STRING database. A**: Validation of co-occurring effector pairs. Effector pairs are sorted by their co-occurring frequencies in Super Modules, and their cumulative STRING scores are calculated. The ranks of the sorted pairs (x axis) and their cumulative STRING scores (y axis) are displayed. **B**: Validation of co-occurring effector-target pairs. Legend follows panel **A**.
(TIF)

**S16 Fig. Illustration of the algorithm to generate Super Modules and Sample Groups.** The algorithm comprises three phases. In phase 1, Modules and Samples are sorted by recursively incurring spectral clustering to the rows (Modules) and columns (Samples) of the data separately. In each iteration, subunits (Modules or Samples) are partitioned into two groups. The outputs of phase 1 are the expression data with sorted Modules and Samples. In phase 2, the sorted data are smoothened by convolving with Gaussian kernels with varying scales (standard deviations). Boundaries of each smoothened data are detected. In phase 3, boundaries which are robust against varying scales are selected. The Super Modules and Sample Groups are demarcated by the selected boundaries. (TIFF)

**S1 Table. Summary information of integrated hierarchical association structure. A**: Number of each type of Association Modules in each cancer type. **B**: Summary information of 217 Super Modules. **C**: Summary information of 228 Sample Groups. **D**: Effectors and regulators of 217 Super Modules. **E**: Super Module-gene membership matrix with sorted genes and Super Modules according to Fig 3A. **F**: Coarse-grained over-representation matrix of Gene Groups in Super Module Groups. (XLSX)

**S2 Table. Gene Set enrichment outcomes of integrated hierarchical association structure. A**: FDR-adjusted Gene Set enrichment p-values of Gene Groups. **B**: FDR-adjusted Gene Set enrichment p-values of Super Modules. **C**: FDR-adjusted p-values of the Gene Sets which are uniquely enriched in each Super Module. (XLSX)

**S3 Table. Complete Recurrent Effectors of all Super Module Groups.** (XLSX)

**S4 Table. Consensus Artery Network information of integrated hierarchical association structure. A**: Consensus Artery Network. **B**: Levels and degrees of nodes in the Consensus Artery Network. **C**: Enriched Gene Sets and pathways in the Consensus Artery Network. (XLSX)

**S5 Table. Information of Sample Groups and their alignments with clinical features. A**: Sample Group labels of all samples in the TCGA data. **B**: Concentration coefficients of aligning Sample Groups with clinical features. **C**: Alignments of Sample Groups with selected clinical features in four cancer types displayed in S4 Fig. **D**: The selected clinical feature values of the sorted samples in four cancer types displayed in S4 Fig. **E**: Pan-cancer Sample Group assignments of Sample Groups. **F**: Sorted samples appeared in Fig 4B. (XLSX)

**S6 Table. Prognostic analysis outcomes of integrated hierarchical association structure. A**: Deviation scores of Cox regression coefficient distributions and log rank p-values of Kaplan-Meier curves for Super Modules. **B**: The same scores for Gene Groups in each cancer type. (XLSX)

**S7 Table. Validation outcomes on METABRIC data. A**: mRNA expression coherence scores of breast cancer Super Modules in TCGA BRCA and METABRIC data. **B**: CNV-mRNA expression association strengths of breast cancer Super Modules in TCGA BRCA and METABRIC data. **C**: Deviation scores of Cox regression coefficient distributions and log rank p-values of Kaplan-Meier curves for breast cancer Super Modules in TCGA BRCA and METABRIC data. (XLSX)

**S8 Table. Validation outcomes on REMBRANDT data.** Legend follows S7 Table.
(XLSX)

**S9 Table. Summary information of GEO datasets and their validation outcomes.**
(XLSX)

**S10 Table. Validation outcomes on CCLE and Achilles data. A**: mRNA expression coherence scores of Gene Groups. Calculation results based on all cancer cell lines and the cancer cell lines derived from the 33 cancer types in TCGA are reported. **B**: CNV-mRNA expression expression association strengths. **C**: Recurrent mutation-mRNA expression association strengths. **D**: Recurrent DNA methylation-mRNA expression association strengths. **E**: Recurrent microRNA expression-mRNA expression association strengths. **F**: Recurrent protein phosphorylation-mRNA expression association strengths. **G**: Drug response-mRNA expression association strengths. **H**: Sorted perturbed genes in Achilles data that appear in Fig 9C and 9D. **I**: Gene Set enrichments of Achilles data gene clusters that appear in Fig 9C and 9D. **J**: Sorted candidate drug target genes in each perturbed gene cluster.
(XLSX)

**S11 Table. Validation outcomes on Illumina Bodymap data. A**: mRNA expression coherence of Gene Groups. **B**: 8494 genes which are uniquely expressed in each of 16 normal tissues. **C**: Enrichment p-values of tissue-specific genes in sorted Super Modules appeared in Fig 11A. **D**: Enrichment p-values of tissue-specific genes in Gene Groups.
(XLSX)

**S12 Table. Validation outcomes on Roadmap epigenomic data. A**: 25 epigenomic states in the data. **B**: Coherence of 18 Gene Groups on binary epigenomic states. **C** Members of 8 gene clusters. **D**: Enrichment p-values of sorted Super Modules appeared in Fig 11B in Roadmap gene clusters. **E**: Enrichment p-values of Gene Groups in Roadmap gene clusters.
(XLSX)

**S13 Table. Comparison of IHAS with several multi-omics, pan-cancer studies. A**: Presence or absence of 12 features in IHAS and 5 other methods. **B**: Overlap counts and enrichment p-values of 8 IHAS Pan-cancer Sample Groups with 28 sample iClusters. **C**: Overlap counts and enrichment p-values of 8 IHAS Pan-cancer Sample Groups with 6 immune subtypes. **D**: Overlap counts and enrichment p-values of 18 IHAS Gene Groups with 24 MOMA MRBs. **E**: Overlap counts and enrichment p-values of 18 IHAS Gene Groups with 29 TME gene signatures. **F**: Overlap counts and enrichment p-values of 8 IHAS Pan-cancer Sample Groups with 4 TME subtypes. **G**: Alignment outcomes between IHAS Sample Groups and MOFA sample clusters in each cancer type. **H**: Comparison of co-occurring Effector pairs and the STRING database of protein-protein associations. **I**: Comparison of the recurrent effector-target pairs and the STRING database.
(XLSX)

**S1 Text. A detailed description of data processing and analysis methods and some analysis results in the paper.**
(PDF)

## Acknowledgments

We thank TCGA Consortium and METABRIC Consortium for granting us to access the multi-omics cancer data. We also thank comments from Hsueh-Chi Yen and Pei-Ing Huang about the manuscript.

## Author Contributions

**Conceptualization:** Chen-Hsiang Yeang.

**Data curation:** Khong-Loon Tiong, Nardnisa Sintupisut, Min-Chin Lin, Chih-Hung Cheng, Andrew Woolston, Chih-Hsu Lin, Mirrian Ho.

**Formal analysis:** Andrew Woolston, Chen-Hsiang Yeang.

**Funding acquisition:** Chen-Hsiang Yeang.

**Investigation:** Khong-Loon Tiong, Andrew Woolston, Chen-Hsiang Yeang.

**Methodology:** Khong-Loon Tiong, Andrew Woolston, Chen-Hsiang Yeang.

**Project administration:** Chen-Hsiang Yeang.

**Resources:** Khong-Loon Tiong, Nardnisa Sintupisut, Chih-Hsu Lin, Mirrian Ho.

**Software:** Khong-Loon Tiong, Chen-Hsiang Yeang.

**Supervision:** Chen-Hsiang Yeang.

**Validation:** Khong-Loon Tiong, Min-Chin Lin, Chih-Hung Cheng, Yu-Wei Lin, Sridevi Padakanti, Chen-Hsiang Yeang.

**Visualization:** Khong-Loon Tiong, Chen-Hsiang Yeang.

**Writing – original draft:** Khong-Loon Tiong, Chen-Hsiang Yeang.

**Writing – review & editing:** Khong-Loon Tiong, Chen-Hsiang Yeang.

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
