## [Decision Letter · Decision Letter 0]

22 Jun 2022

PDIG-D-22-00162

An integrated analysis of the cancer genome atlas data discovers a hierarchical association structure across thirty three cancer types

PLOS Digital Health

Dear Dr. Yeang,

Thank you for submitting your manuscript to PLOS Digital Health. After careful consideration, we feel that it has merit but does not fully meet PLOS Digital Health's publication criteria as it currently stands. Therefore, we invite you to submit a revised version of the manuscript that addresses the points raised during the review process.

All reviewers recommended to revise the manuscript in a readable length while addressing various concerns on the rationales, methodology and data interpretation of this study. Detailed comments from reviewers can be found below.

In particular, cancer is highly individual- and organ-specific. Please justify how the proposed global analysis will address patient/ organ-specific cancer manifestation and response to treatments, rather than making general and broad statements across multiple cancers.

Please submit your revised manuscript within 60 days . If you will need more time than this to complete your revisions, please reply to this message or contact the journal office at digitalhealth@plos.org. Please include the following items when submitting your revised manuscript:

We look forward to receiving your revised manuscript.

Kind regards,

Nicole Yee-Key Li-Jessen

Academic Editor

PLOS Digital Health

Journal Requirements:

a. Please clarify all sources of funding (financial or material support) for your study. List the grants (with grant number) or organizations (with url) that supported your study, including funding received from your institution. 

b. State the initials, alongside each funding source, of each author to receive each grant.

c. State what role the funders took in the study. If the funders had no role in your study, please state: “The funders had no role in study design, data collection and analysis, decision to publish, or preparation of the manuscript.”

d. If any authors received a salary from any of your funders, please state which authors and which funders.

2. Please ensure that the funders and grant numbers match between the Financial Disclosure field and the Funding Information tab in your submission form. Note that the funders must be provided in the same order in both places as well.

3. Please update your online Competing Interests statement. If you have no competing interests to declare, please state: “The authors have declared that no competing interests exist.”

4. We ask that a manuscript source file is provided at Revision. Please upload your manuscript file as a .doc, .docx, or .rtf.

5. Please provide separate figure files in .tif or .eps format and remove any figures embedded in your manuscript file. Please also ensure that all files are under our size limit of 10MB.

For more information about how to convert your figure files please see our guidelines: https://journals.plos.org/digitalhealth/s/figures

Additional Editor Comments (if provided):

Reviewers' comments:

Reviewer's Responses to Questions

**Comments to the Author**

1. Does this manuscript meet PLOS Digital Health’s publication criteria? Is the manuscript technically sound, and do the data support the conclusions? The manuscript must describe methodologically and ethically rigorous research with conclusions that are appropriately drawn based on the data presented.

Reviewer #1: Partly

Reviewer #2: Yes

Reviewer #3: No

Reviewer #4: Yes

2. Has the statistical analysis been performed appropriately and rigorously?

Reviewer #1: N/A

Reviewer #2: Yes

Reviewer #3: No

Reviewer #4: N/A

3. Have the authors made all data underlying the findings in their manuscript fully available (please refer to the Data Availability Statement at the start of the manuscript PDF file)?

Reviewer #1: No

Reviewer #2: Yes

Reviewer #3: No

Reviewer #4: Yes

4. Is the manuscript presented in an intelligible fashion and written in standard English?

Reviewer #1: Yes

Reviewer #2: Yes

Reviewer #3: Yes

Reviewer #4: Yes

5. Review Comments to the Author

Reviewer #1: The manuscript PDIG-D-22-00162 entitled “An integrated analysis of the cancer genome atlas data discovers a hierarchical association structure across thirty three cancer types” produced an Integrated Hierarchical Association Structure (IHAS) from the complete data of TCGA and compiled a large database of cancer multi-omics associations.

It is a related work in the fields of cancer omics data study. However, some issues should be addressed well before publication.

(1) The targeted clinical problem is not clear, thus, readers would be not clear how to use such outcomes from this work.

(2) Authors stated that current existing methods would not organize the associations in a hierarchical structure. It is necessary to display the benefit of hierarchical structure. Indeed, the current results have not shown the improvement from hierarchical structure (e.g. Figure 3) and extensive external data (e.g. Figure 4). They are still general heatmap structure and visualization like conventional method, thus there are not novel information produced or available.

(3) In fig8. The validation on GSE data is not significant. There should be lots of evidences with significant biological or biomedical meaning.

(4) In particular, this is a work on multi-omics in pan-cancer, however, there are not enough results and discussions on the contribution from multi-omics analysis, e.g. shared or complementary information from different biological levels.

(5) There are not definite calculation model and measurements in main text. I only saw descriptions of concepts. More solid details of different so-called modules should supply their analysis and biology hypothesis.

(6) The key is the database and useful web-server for experts from different fields. But, in current form, there is not detail database and web introduced and discussed in main text. It is necessary to provide the complete database for public domain.

Reviewer #2: In this paper, the authors present a pan-cancer analysis of TCGA data for hierarchical association structure across multiple cancers. The major concern is what are the main findings from the integrative data analysis. What are their indicators for cancer studies? How to justify these findings? This paper provides a resource of building the complicated associations in a hierarchical structure (across multiple data types and across multiple cancer types). It is expected to clarify some concrete conclusions for this hierarchical structure. As for so many cancer types, it is good to present the common features in these associations. For cancer specific features, it is also suggested to present in a rational way. This paper is very long and it is hard to grasp the main idea for the reader’s perspective. It is possible to summarize the main findings more clearly. The cancer types are diverse as described. Is it possible to cluster them into several groups and summarized the common features in the hierarchical structures. It is good to compare the present method to the other omics integration methods. How to evaluate the results by different data integration methods? Thus, the authors need present their work more clearly.

Reviewer #3: This paper studies the hierarchical correlation structure of 33 cancer types, but there are problems such as a lot of repetition and no emphasis. In addition, there are also big problems in the schematic diagram, as follows:

1.The introduction contained so many descriptions about previous studies, and lacked comparisons and conclusions.

2.In the results section, the biological explanations were missing, and many concepts were mentioned so many times. The core conclusions were also missing.

3.There are so small front label for Figure 1, and the workflow of overall design was confusing.

4.The Figure 2 should be totally revised and the label was too small.

5.There were so many examples and no biological evaluations.

6.The authors must provided the website or link for users, which was important for research.

7.Some parts appear many times in the text, such as "three aspects of molecular changes in cancer", suggesting to remove unnecessary parts.

Reviewer #4: In this study, Tiong et al. built an Integrated Hierarchical Association Structure (IHAS) between molecular alterations on genomes/epigenomes and variations on transcriptomes in 33 cancer types. They justified the biological relevance and clinical utility of IHAS by characterizing its functional properties, aligning combinatorial expressions of IHAS subunits with phenotypes, and validating IHAS in a wide range of external datasets. IHAS seems valid and the results they show are promising. However, there are existing some specific problems.

1. I did not see the Web page of IHAS. Is it possible to use IHAS through its web interface? Please make it publicly accessible.

2. The manuscript is too long, and it is not easy to follow the information disclosed by IHAS. The authors may split it into two (one is about IHAS framework and the other is the discovery from IHAS).

3. Besides validated IHAS in more than 300 external datasets, could the authors validate several discoveries using biomedical experiment?

6. PLOS authors have the option to publish the peer review history of their article (what does this mean?). If published, this will include your full peer review and any attached files.

**Do you want your identity to be public for this peer review?** For information about this choice, including consent withdrawal, please see our Privacy Policy.

Reviewer #1: No

Reviewer #2: No

Reviewer #3: No

Reviewer #4: No

---

## [Decision Letter · Decision Letter 1]

28 Sep 2022

PDIG-D-22-00162R1

An integrated analysis of the cancer genome atlas data discovers a hierarchical association structure across thirty three cancer types

PLOS Digital Health

Dear Dr. Yeang,

Thank you for submitting your manuscript to PLOS Digital Health. After careful consideration, we feel that it has merit but does not fully meet PLOS Digital Health's publication criteria as it currently stands. Therefore, we invite you to submit a revised version of the manuscript that addresses the points raised during the review process.

The revised manuscript has largely addressed the reviewers' comments. However, there are still concerns on the figures, especially Figure 1. Please refer to the detailed comments from the reviewers.

Please submit your revised manuscript within 30 days Oct 28 2022 11:59PM. If you will need more time than this to complete your revisions, please reply to this message or contact the journal office at digitalhealth@plos.org. Please include the following items when submitting your revised manuscript:

We look forward to receiving your revised manuscript.

Kind regards,

Nicole Yee-Key Li-Jessen

Academic Editor

PLOS Digital Health

Journal Requirements:

2. Please insert an Ethics Statement at the beginning of your Methods section, under a subheading 'Ethics Statement'. It must include:

a) The name(s) of the Institutional Review Board(s) or Ethics Committee(s)

b) The approval number(s), or a statement that approval was granted by the named board(s) 

c) (for human participants/donors) - A statement that formal consent was obtained (must state whether verbal/written) OR the reason consent was not obtained (e.g. anonymity). NOTE: If child participants, the statement must declare that formal consent was obtained from the parent/guardian.

Additional Editor Comments (if provided):

Reviewers' comments:

Reviewer's Responses to Questions

**Comments to the Author**

1. If the authors have adequately addressed your comments raised in a previous round of review and you feel that this manuscript is now acceptable for publication, you may indicate that here to bypass the “Comments to the Author” section, enter your conflict of interest statement in the “Confidential to Editor” section, and submit your "Accept" recommendation.

Reviewer #1: (No Response)

Reviewer #2: All comments have been addressed

Reviewer #4: All comments have been addressed

2. Does this manuscript meet PLOS Digital Health’s publication criteria? Is the manuscript technically sound, and do the data support the conclusions? The manuscript must describe methodologically and ethically rigorous research with conclusions that are appropriately drawn based on the data presented.

Reviewer #1: Yes

Reviewer #2: Yes

Reviewer #4: Yes

3. Has the statistical analysis been performed appropriately and rigorously?

Reviewer #1: N/A

Reviewer #2: Yes

Reviewer #4: N/A

4. Have the authors made all data underlying the findings in their manuscript fully available (please refer to the Data Availability Statement at the start of the manuscript PDF file)?

Reviewer #1: Yes

Reviewer #2: Yes

Reviewer #4: Yes

5. Is the manuscript presented in an intelligible fashion and written in standard English?

Reviewer #1: Yes

Reviewer #2: Yes

Reviewer #4: Yes

6. Review Comments to the Author

Reviewer #1: Authors have responded to all my concerned questions. There is only few remaining issue.

For the question about “details of different so-called modules should supply their analysis and biology hypothesis”, authors replied “Since the length of the paper is already criticized by most reviewers, we think supplying more technical information in the main text will probably raise more criticism and confuse readers.” 

Indeed, IHAS is not a general model, thus, the current main figure and text didn’t provide enough information for me and readers in different fields. The current figure 1 is too simple to start the paper. In current form, Figure 1 should, at least, supply a clear demonstration of the key characteristics of IHAS, including: the hierarchical association structure (e.g. in a general tree format), the expression pattern corresponding to different module types, the association between modules and clinical indices (e.g. survival), and the shared or complementary expression pattern of one module in different omics data. Of course, a few key definition and formula of these characteristics can also be shown in Figure 1.

Reviewer #2: The authors have addressed my comments. A minor comment is about the figures. They are not clear and hard to recognize. The method section need contain more introduction to the multi-level/hierarchical issues in data integration.

Reviewer #4: The authors have addressed my comments.

7. PLOS authors have the option to publish the peer review history of their article (what does this mean?). If published, this will include your full peer review and any attached files.

**Do you want your identity to be public for this peer review?** For information about this choice, including consent withdrawal, please see our Privacy Policy.

Reviewer #1: No

Reviewer #2: No

Reviewer #4: Yes: Junfeng Xia

---

## [Decision Letter · Decision Letter 2]

31 Oct 2022

An integrated analysis of the cancer genome atlas data discovers a hierarchical association structure across thirty three cancer types

PDIG-D-22-00162R2

Dear Dr. Yeang,

We are pleased to inform you that your manuscript 'An integrated analysis of the cancer genome atlas data discovers a hierarchical association structure across thirty three cancer types' has been provisionally accepted for publication in PLOS Digital Health.

Best regards,

Nicole Yee-Key Li-Jessen

Academic Editor

PLOS Digital Health

Reviewer Comments (if any, and for reference):

Reviewer's Responses to Questions

**Comments to the Author**

1. If the authors have adequately addressed your comments raised in a previous round of review and you feel that this manuscript is now acceptable for publication, you may indicate that here to bypass the “Comments to the Author” section, enter your conflict of interest statement in the “Confidential to Editor” section, and submit your "Accept" recommendation.

Reviewer #1: All comments have been addressed

Reviewer #2: All comments have been addressed

2. Does this manuscript meet PLOS Digital Health’s publication criteria? Is the manuscript technically sound, and do the data support the conclusions? The manuscript must describe methodologically and ethically rigorous research with conclusions that are appropriately drawn based on the data presented.

Reviewer #1: Yes

Reviewer #2: Yes

3. Has the statistical analysis been performed appropriately and rigorously?

Reviewer #1: N/A

Reviewer #2: Yes

4. Have the authors made all data underlying the findings in their manuscript fully available (please refer to the Data Availability Statement at the start of the manuscript PDF file)?

Reviewer #1: Yes

Reviewer #2: Yes

5. Is the manuscript presented in an intelligible fashion and written in standard English?

Reviewer #1: Yes

Reviewer #2: Yes

6. Review Comments to the Author

Reviewer #1: Authors have responded to my concerned question, and made a good revision.

Reviewer #2: My comments have been addressed.

7. PLOS authors have the option to publish the peer review history of their article (what does this mean?). If published, this will include your full peer review and any attached files.

**Do you want your identity to be public for this peer review?** For information about this choice, including consent withdrawal, please see our Privacy Policy.

Reviewer #1: No

Reviewer #2: No
